# Single virus fingerprinting by widefield interferometric defocus-enhanced mid-infrared photothermal microscopy

Qing Xia[1], Zhongyue Guo[2], Haonan Zong[1], Scott Seitz[3], Celalettin Yurdakul [1], M. Selim Ünlü[1], Le Wang[1], John H. Connor [3] ✉ & Ji-Xin Cheng [1,2,4] ✉

Clinical identification and fundamental study of viruses rely on the detection of viral proteins or viral nucleic acids. Yet, amplification-based and antigen-based methods are not able to provide precise compositional information of individual virions due to small particle size and low-abundance chemical contents (e.g., ~ 5000 proteins in a vesicular stomatitis virus). Here, we report a widefield interferometric defocus-enhanced mid-infrared photothermal (WIDE-MIP) microscope for high-throughput fingerprinting of single viruses. With the identification of feature absorption peaks, WIDE-MIP reveals the contents of viral proteins and nucleic acids in single DNA vaccinia viruses and RNA vesicular stomatitis viruses. Different nucleic acid signatures of thymine and uracil residue vibrations are obtained to differentiate DNA and RNA viruses. WIDE-MIP imaging further reveals an enriched β sheet components in DNA varicella-zoster virus proteins. Together, these advances open a new avenue for compositional analysis of viral vectors and elucidating protein function in an assembled virion.

The emergence of the monkeypox outbreak in early 2022 has posed a new global health threat during the coronavirus-19 (COVID-19) pandemic[1–3]. With the spread of virus-based infectious diseases, rapid and accurate testing is crucial for mitigating the impact of current and future pandemics[4]. Diagnostic tests on the viruses commonly rely on the detection of nucleic acids or surface proteins. Generally, the amount of viral nucleic acid in a single virion is lower than the amount of viral protein. Detecting viral nucleic acids is challenging without signal amplification techniques such as polymerase chain reaction. Although nucleic acid amplification tests[4–6] and antigen rapid diagnostic tests[7,8] can provide accurate testing results, they usually require pre-treatments of a large number of virions, extraction, or tagging that add time to any assay[5]. It is noteworthy that residual viral RNA from patient specimens remains detectable even though patients have recovered or without culturable viruses[9–11]. Thus, in addition to detecting viral fragments, new complement assays are required to identify the intact virions with preserved structures in order to confirm viral infection and reduce false diagnoses.

Accelerated efforts have been devoted to developing label-free technologies, in which optical detection and morphological characterization of single viruses have shown to be promising for clinical diagnosis[12,13]. Although the scattering from a single virion is weak, it can be enhanced by interfering with a strong reference field in an interferometric light microscope[14]. With the enhanced signal contrasts, interferometric imaging has been used for single virus tracking and viral infection study[15–17]. Towards translation into the clinic, interferometric sensing methods have also demonstrated the visualization of single viruses in undiluted fetal bovine serum[18] and the rapid detection of a single intact virion in human saliva[19]. However, these methods lack molecular information about the viruses while the chemical contents are critical to viral structure and function[20].

[1]Department of Electrical and Computer Engineering, Boston University, Boston, MA 02215, USA. [2]Department of Biomedical Engineering, Boston University, Boston, MA 02215, USA. [3]Department of Microbiology and National Infectious Diseases Laboratories, Boston University School of Medicine, Boston, MA 02118, USA. [4]Photonics Center, Boston University, Boston, MA 02215, USA. ✉e-mail: jhconnor@bu.edu; jxcheng@bu.edu

Vibrational spectroscopic detection of viruses is valuable for analyzing the chemical components of virus strains[21–23]. Methods relying on either Raman scattering or infrared (IR) absorption offer intrinsic chemical selectivity at a single virus level by using spectroscopic signatures of chemical bonds[24,25]. Compared to Raman scattering, IR absorption offers 8 orders of magnitude larger cross-section that enables adequate chemical sensitivity and throughput[26]. Mid-infrared photothermal (MIP) microscopy is an emerging technique based on the mapping of local transient heat to achieve IR spectroscopic imaging at the diffraction limit of visible light[27,28]. In MIP microscopy, a visible probe beam is used to detect photothermal-based chemical contrast induced by a mid-IR pump beam[29]. Since the first demonstration of 3D MIP imaging of living cells[29], MIP microscopy has enabled broad applications in life science, ranging from individual bacteria[30], single cells[31–33], sliced tissues[34], to entire organisms[35]. With counter-propagation of IR and visible beams, researchers have shown MIP imaging of 100 nm polystyrene beads[36,37]. With interferometric scattering as the probe in a confocal configuration, MIP spectroscopic detection of a single virus was reported[38]. However, the scanning-based MIP methods suffer from long acquisition time and low throughput. Although widefield MIP imaging was developed to allow ultrafast chemical imaging at a speed up to 1250 frames per second[39], it remains very challenging for widefield MIP to detect single viral nanoparticles and perform precise spectral analysis.

Here, we present the development and validation of a widefield interferometric defocus-enhanced MIP (WIDE-MIP) microscope (Fig. 1a) for fingerprint analysis of bionanoparticles. As photothermal signal is a modulation of visible beam intensity, it is commonly believed that an optimal MIP contrast is generated when the bright field contrast is maximized. Yet, this wisdom does not hold for interferometric MIP microscopy where the signal strongly depends on the relative phase between the particle-scattered photons and the substrate-reflected reference field. Instead, we find that by fine-tuning the focus position of the objective, the defocused interferometric imaging results in a greatly improved MIP contrast (Fig. 1b). We constructed a theoretical framework that calculates the defocused interferometric photothermal images of single nanoparticles with different sizes. This framework provides the optimal MIP detection focus position relative to the nominal focus position of the particles. Compared to reported scanning methods[36–38], we demonstrate vibrational detection of 100 nm polymethyl methacrylate (PMMA) particles at a similar signal-to-noise ratio (SNR) level but with three orders of magnitude higher throughput. By tuning the IR wavenumber, WIDE-MIP spectra of single viruses are acquired from the hyperspectral images (Fig. 1c). We systematically recorded the fingerprints of single vaccinia viruses (VACV), a DNA poxvirus related to Monkeypox[40], single vesicular stomatitis viruses (VSV), the prototype RNA virus[41], and varicella-zoster viruses (VZV), a DNA virus included in the herpesvirus group[42]. Dramatically, the spectra provide signatures of not only viral proteins, but also nucleic acids of individual viruses. Nucleic acid peaks of thymine (T) and uracil (U) residue vibrations in VACV and VSV were detected respectively, indicating unique IR signatures of DNA and RNA viruses. Besides the contents, WIDE-MIP data further suggests a β-enriched sheet structure in VZV, showing the potential of analyzing protein secondary structure in a single virus.

## Results

### Theory and experimental validation of WIDE-MIP detection

WIDE-MIP is a highly sensitive vibrational detection platform based on infrared photothermal modulation of interferometric scattering. The schematic of the WIDE-MIP microscope is illustrated in Fig. 1a. In previous implementation of widefield MIP[39], a visible LED was utilized as the probe light, which had a relatively long pulse duration of ~1 μs and only allowed detection of PMMA beads of 1.0 μm diameter. To match the nanosecond-scale thermal decay of nanoparticles (240 ns for 200-nm PMMA beads in air[28]), we incorporated a nanosecond pulsed laser (NPL52C, Thorlabs, pulse duration of 129 ns) as the visible probe to improve the sensitivity. A pulsed mid-infrared laser (Firefly-LW, M Squared Lasers) excites the sample placed on a silicon substrate. The visible probe $E_i$ illuminates the sample and is further scattered by the sample $E_s$ and reflected by the substrate $E_r$. Compared to a transparent substrate, such as calcium fluoride, silicon reflects most of the forward-scattered light and increases the total back-scattering[43]. Consequently, the scattered light interfered with the reflected light and the resulting interferometric image represents the coherent sum of the scattered and reflected fields[43,44]:

$$I_{det} = |E_r + E_s|^2 = |E_r|^2 + |E_s|^2 + 2|E_r||E_s|\cos\varphi \qquad (1)$$

where $\varphi$ is the phase difference between $E_s$ and $E_r$. The normalized interferometric contrast $S$ is defined as:

$$S = \frac{I_{det-bkg}}{I_{bkg}} = \frac{|E_r + E_s|^2 - |E_r|^2}{|E_r|^2} = \frac{|E_s|^2}{|E_r|^2} + 2\frac{|E_s|}{|E_r|}\cos\varphi \qquad (2)$$

where $I_{bkg}$ is the background intensity.

For particles of small size like viruses, $|E_r|^2 \gg |E_s|^2$. Then, we have

$$S \cong 2\frac{|E_s|}{|E_r|}\cos\varphi \qquad (3)$$

The photothermal contrast $C$ induced by IR absorption is generated from the interferometric contrast difference between IR-on (hot)

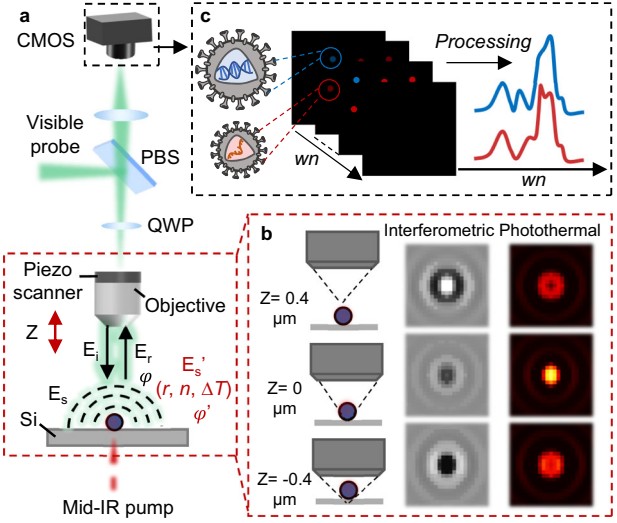

**Fig. 1 | Schematic and principle of WIDE-MIP microscopy. a** Schematic of a WIDE-MIP microscope and principle of interferometric scattering-based MIP imaging. CMOS complementary metal-oxide semiconductor, PBS Polarizing beam splitter, QWP quarter-wave plate, $E_i$ visible incident light field, $E_r$ reflected field by the substrate, $E_s$ scattered field by the sample, $E_s$' IR modulation resulted scattered field by the sample, which is related to the radius ($r$), refractive index ($n$), and temperature change ($\Delta T$) of the sample, $\varphi$ phase difference between $E_s$ and $E_r$. A delay pulse generator is used to synchronize the pump pulse, probe pulse, and camera (Supplementary Fig. 1). **b** Schematic illustration of interferometric defocus-enhanced photothermal contrast provided by Z axis scanning of objective. The different positions are with respect to the substrate top surface ($Z = 0 \mu m$). **c** Principle of fingerprinting DNA and RNA viruses by WIDE-MIP. wn wavenumber. Hyperspectral images of single viruses are recorded by continuously tuning the IR wavenumber. Blue dots indicate DNA viruses, red dots indicate RNA viruses.

and IR off (cold) states:

$$C = \frac{2}{|E_r|}(E_s^{hot} \cos\varphi^{hot} - E_s^{cold} \cos\varphi^{cold}) \qquad (4)$$

where $|E_r|$ is assumed as a constant in the modulation. For the purpose of brevity, only the change of $E_s$ is taken into account between hot and cold states and $\varphi$ is considered as a constant in previous MIP work[38,45,46]. However, for specular reflection, $E_r$ only travels in one direction and is reflected back along the optical axis, while $E_s$ travels in all directions, mostly at oblique angles. Due to thermal expansion of the particle, the traveling direction of $E_r$ relative to $E_s$ is different, and thus the phase angle $\varphi$ is slightly different in hot and cold states. Because the phase angle $\varphi$ also depends on the axial position of the optical focus, the MIP contrast can be optimized by tuning the focal position. To precisely control the $Z$ axis scanning, the objective is mounted on an objective piezo scanner for defocus-enhanced photothermal image acquisition (Fig. 1b).

To validate the interferometric phase difference, the interferometric image of a 200 nm diameter (D) PMMA bead was numerically simulated via the boundary element method (BEM)[47]. Interferometric contrast $S$ is then calculated using the metallic nanoparticle boundary element method (MNPBEM) toolbox[45]. The MIP signal is generated from the interferometric contrast difference between IR-on (hot) and IR off (cold) states. The transient temperature difference between hot and cold states is set to be ~80 K over a temporal window of 129 nanoseconds (duration of probe pulse), which is calculated from COMSOL simulation[36,45] (Supplementary Fig. 2, details in Supplementary Note 1). We simulated the interferometric images of the 200 nm PMMA bead at both cold ($T = 293.15$ K) and hot ($T = 373.15$ K) states along the $Z$ axis focus of the objective. The interferometric contrast at the center of the diffraction-limited image of the PMMA bead on a silicon substrate is calculated as the focus position $Z$ is swept. Here, $Z$ is set to be zero for exact optical focusing at the sample-substrate interface for the light-collecting objective. As shown in Fig. 2a, the simulated defocus curves of cold and hot contrasts have a similar sinusoidal function shape, both of them reaching the maximum contrast near $Z = 0.4$ μm. For the hot state, the increased local temperature changes the opto-physical properties of the PMMA particle, such as size ($r$) and refractive index ($n$). As seen from the zoomed-in view at different focal planes, the slopes of the interferometric contrast vary greatly as a function of $Z$ (Fig. 2b, c). Strikingly, the interferometric contrast is least sensitive to the axial focus when the particle contrast is maximized at $Z = 0.4$ μm[15]. Consequently, the photothermal contrast is only 0.00035% at $Z = 0.4$ μm (Fig. 2c). On the contrary, the interferometric contrast is most sensitive to the change in opto-physical properties of the particle caused by the local temperature increase near the interface at $Z = 0$ μm, where the photothermal contrast is ~0.6% at $\Delta T = 80$ K (Fig. 2b). Therefore, the difference between cold and hot state, defined as the MIP contrast, is maximized at a defocused plane relative to the interferometric contrast. The interferometric image shows a bright contrast at the focal plane of $Z = 0.4$ μm, where the MIP contrast is low. The MIP image reaches its maximum contrast at $Z = 0$ μm, where the interferometric image shows a negative contrast (Fig. 2d, e).

To experimentally validate the mechanism of interferometric defocus-enhanced MIP, $D = 200$ nm PMMA beads on a silicon substrate were used. We first used interferometric contrast (IR off) to locate the beads under the microscope. Once the beads were observed in the focal plane, the IR laser was turned on to 1728 cm⁻¹, which corresponds to the acrylate carboxyl vibration (C=O stretching) in PMMA. To optimize the MIP contrast, we manually adjusted the defocus with a piezo scanner. Subsequently, a series of interferometric and MIP images of the PMMA beads were acquired by scanning the focal

position of the objective lens. As shown in Fig. 2f, g, the experimental images match the simulation results very well.

To derive the optimal condition for MIP imaging, we plotted the interferometric and MIP contrast as a function of optical focus position through both simulation (Fig. 2h) and experiment (Fig. 2i). The simulated focal plane difference between interferometric and MIP images, $\Delta Z = 400$ nm, is highly consistent with the experimental result. By defocusing the interferometric images, the MIP contrast is increased by 2.5 times for 200 nm PMMA particles. For PMMA beads with different sizes, the defocus curve and $\Delta Z$ have different shapes and values (Supplementary Fig. 3, details in Supplementary Note 2). Thus, this framework provides a guideline to obtain a well-defined and optimized photothermal detection signal by adjusting the focus for MIP detection of nanoparticles with different sizes.

## Hyperspectral performance and spectral fidelity

To further test the capability of WIDE-MIP for spectroscopic imaging of single virus in the fingerprint window, we evaluated the hyperspectral performance and spectral fidelity of the system. Single 200-nm PMMA beads with known IR absorption spectrum were chosen for their similar size and dielectric constant ($n \approx 1.5$) to the Monkeypox viruses. Figure 3a, b show MIP images of the beads at 1452 cm⁻¹ and 1728 cm⁻¹, indicating bond-selective contrast from the C−H and C=O stretching of PMMA. The statistical spectra of 30 individual beads showed the distinguished resonance peaks of both C−H and C=O stretching vibrations (Fig. 3c, red line). The standard deviation of the mean MIP contrast within the range of ~1510 to 1610 cm⁻¹ was found to be ~0.16%, which corresponds to the off-resonance region of PMMA vibration. This demonstrates the stable hyperspectral performance of WIDE-MIP. Furthermore, the spectral fidelity was confirmed by comparing the WIDE-MIP spectrum to FTIR absorption spectrum of PMMA (Fig. 3c, black line)[48]. With the increased MIP contrast, WIDE-MIP realizes the high-speed widefield photothermal detection of D = 100 nm PMMA nanoparticles (Supplementary Fig. 4, details in Supplementary Note 3), which increases the throughput by 3 orders of magnitude compared with scanning MIP at a similar level of SNR (Supplementary Table 1)[36–38].

## Fingerprinting and base residue detection of single DNA and RNA viruses

With its high-resolution and high-throughput capability, WIDE-MIP opens the possibility of single virus chemical detection. We used single VACV and VSV viruses as testbeds. The dimensions of the VACV virion are roughly $360 \times 270 \times 250$ nm[49]. VSV is a bullet-shaped RNA virus with a smaller size of $80 \times 180$ nm[38,50,51]. Figure 4a shows the defocused interferometric scattering image of single VACV viruses. It should be noted that the depth of focus for MIP imaging is 503 nm and the spatial resolution is 417 nm, measured from a 200 nm PMMA particle in MIP image captured at the defocus plane of $Z = 0$ μm (Supplementary Fig. 5, details in Supplementary Note 4). To confirm MIP imaging of single viruses, both VACV and VSV viruses were expressed with an enhanced green fluorescent protein (eGFP) envelope for an orthogonal validation. With the good overlay of the widefield fluorescence imaging (Fig. 4b) and the interferometric scattering images (Fig. 4c), we confirmed that the observed particles were indeed VACV virions. Atomic force microscope analysis further confirmed the size of single virions (Supplementary Fig. 6, details in Supplementary Note 5). Bond-selective MIP imaging showed the amide II (1544 cm⁻¹) and amide I (1656 cm⁻¹) vibrational contrasts contributed by viral proteins (Fig. 4d, e), whereas the off-resonance images at 1768 cm⁻¹ showed no contrasts (Fig. 4f). Similar results of single VSV viruses are shown in the defocused interferometric scattering, fluorescence, and MIP images (Fig. 4g–l). As VSV is less concentrated on the imaging plate than VACV due to the preparation procedure, we provided more data in Supplementary Fig. 7.

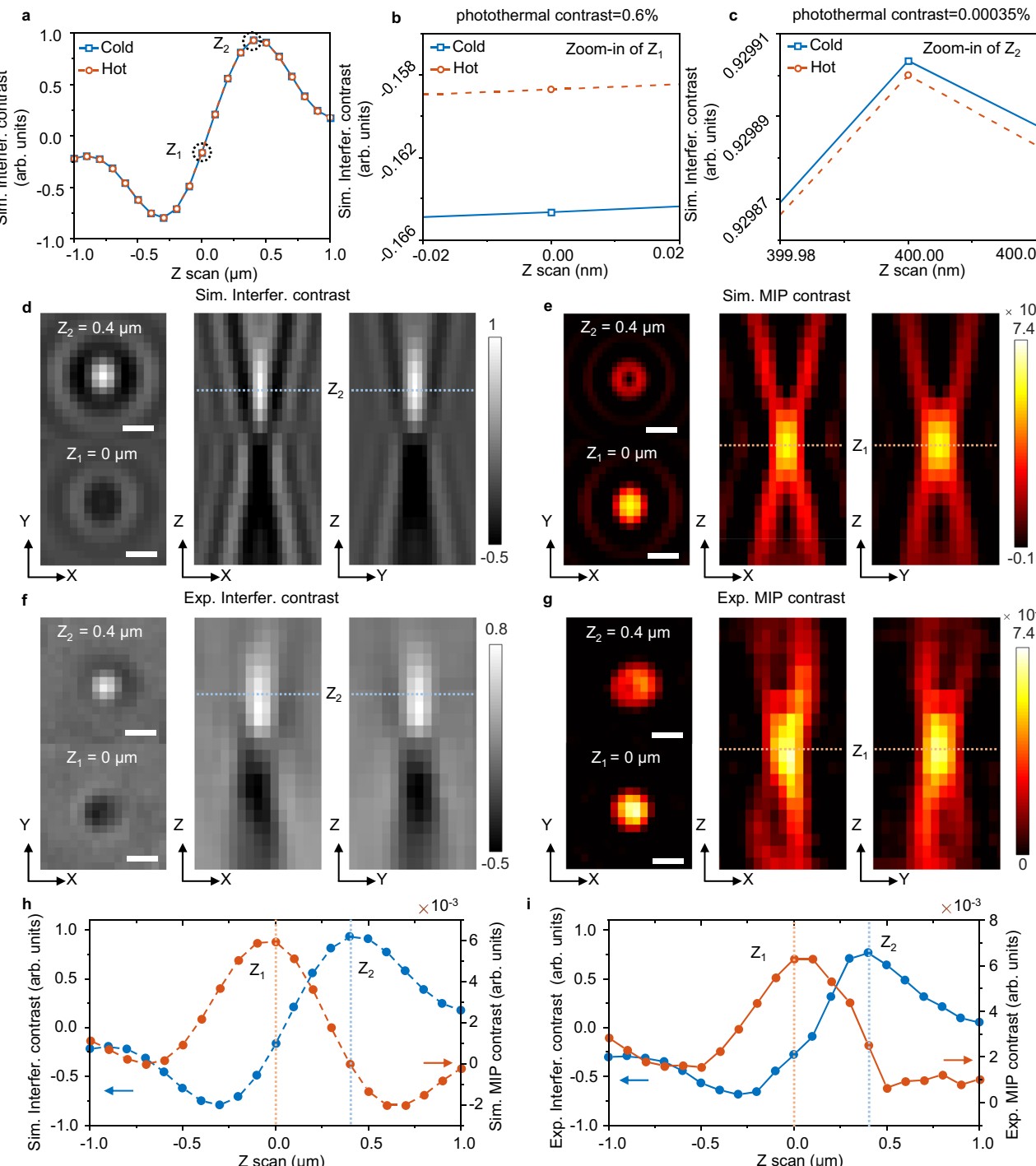

**Fig. 2 | Simulation and experimental validation of interferometric defocus-enhanced photothermal contrast.** PMMA beads of $D = 200$ nm were used as the testbed. **a** Simulated defocus curves of interferometric contrast at the cold ($T = 293.15$ K) and hot ($T = 373.15$ K) states. Zoomed-in-view of simulated defocus curves of interferometric contrast at the position of **b** $Z_1 = 0$ μm and **c** $Z_2 = 0.4$ μm. Interfer.: interferometric. The photothermal contrast is 0.6% at $Z_1 = 0$ μm and 0.00035% at $Z_2 = 0.4$ μm. **d** Simulated interferometric images at $Z_2 = 0.4$ μm, $Z_1 = 0$ μm (left), and interferometric scattering along $Z$ axis (right). **e** Simulated MIP images at $Z_2 = 0.4$ μm, $Z_1 = 0$ μm (left), and MIP imaging along $Z$ axis (right). **f** Experimental interferometric images at $Z_2 = 0.4$ μm, $Z_1 = 0$ μm (left), and interferometric scattering along $Z$ axis (right). **g** Experimental MIP images at $Z_2 = 0.4$ μm, $Z_1 = 0$ μm (left), and MIP imaging along $Z$ axis (right). Scale bar: 500 nm. All $Z$ axis images are obtained from $Z = -1$ to 1 μm. **h** Simulated and **i** experimental defocus curves of interferometric and MIP contrast. Power before the objective: pump: 48 mW at 1728 cm⁻¹, probe: -1 mW. Image acquisition time: 2.36 s per image. $Z$ axis scanning step: 100 nm. Source data are provided as a Source Data file.

To provide further insight into the viral structure and content, we performed WIDE-MIP hyperspectral imaging of single VACV and VSV viruses (blue and red arrows labeled in Fig. 4a–l). Obvious differences were overserved in the single-virus fingerprints (Fig. 4m). Merited from the high-throughput ability of spatial multiplexing of WIDE-MIP,

spectral analysis of multiple viruses was performed. The statistical spectra of both VACV ($n = 36$) and VSV ($n = 33$) (Fig. 4n, o) are in good agreement with the single-virus spectra (Fig. 4m). Besides the viral protein vibrations, some unique peaks reveal the information of the viral nucleic acids. Different from the wide amide I peak from the pure

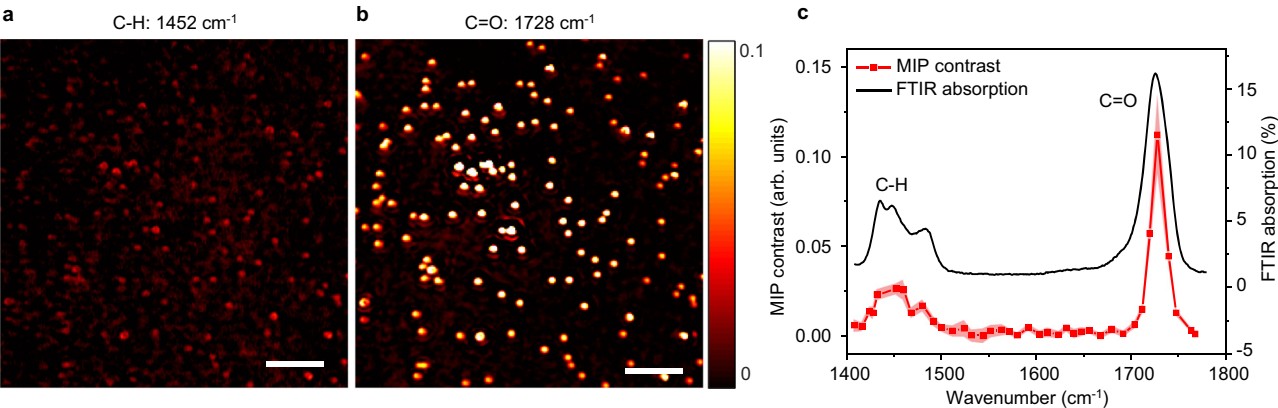

**Fig. 3 | Hyperspectral performance and spectral fidelity of WIDE-MIP microscopy.** MIP image of $D = 200$ nm PMMA beads with IR excitation at **a** 1452 cm$^{-1}$ and **b** 1728 cm$^{-1}$. Scale bars: 5 μm. Experiments were repeated at least three times. **c** MIP spectrum (red) and FTIR spectrum (black) of $D = 200$ nm PMMA beads. $n = 30$ for biologically independent PMMA beads. Error bands represent the standard deviation of the mean. Power before the objective: pump: 31.4 mW at 1452 cm$^{-1}$, 38.6 mW at 1728 cm$^{-1}$, probe: -1 mW. Image acquisition time: 2.36 s per wavenumber. The MIP spectrum was normalized by the IR power. The FTIR spectrum was acquired by an attenuated total reflection FTIR spectrometer. Source data are provided as a Source Data file.

protein samples (Supplementary Fig. 8a), the strongest sharp peak at 1656 cm$^{-1}$ is contributed by a superposition of the viral protein, adenine (A), and T residue vibrations in viral DNA of VACV (Fig. 4n). A medium feature at 1580 cm$^{-1}$ is assigned to the T residue vibration in VACV viral DNA. For VSV, the U residue vibration in RNA is indicated by the strong peak at 1640 cm$^{-1}$ (Fig. 4o). More detailed features of nucleic acids in VSV are revealed by the weak peak at 1604 cm$^{-1}$ (A and cytosine (C)) and strong peak at 1656 cm$^{-1}$ (A and proteins). The guanine (G) residue vibrations are identified at 1692 cm$^{-1}$ in both VACV and VSV. The assignments of the chemical components were validated by the pure protein, DNA, and RNA film samples[52] (Supplementary Fig. 8, details in Supplementary Note 6). It indicates that WIDE-MIP can provide rich chemical content information of viral proteins and even nucleic acids inside a single virus.

To further demonstrate the potential of WIDE-MIP to differentiate single RNA viruses from single DNA viruses, we compared the signature peaks of nucleic acids at the single-virus level by quantifying the MIP contrast of peaks at T residue and U residue for VACV and VSV to highlight the spectroscopic difference of DNA and RNA viruses. Although the MIP contrasts of VACV and VSV show no statistically significant difference at 1580 cm$^{-1}$ representing U residue (Fig. 4p, $P = 0.138892$), MIP contrasts of both T residue (Fig. 4q, $P = 7.4 \times 10^{-5}$) and the ratio of T/U (Fig. 4r, $P = 5.2 \times 10^{-14}$) show significant difference between VACV and VSV. Our results show that fingerprint WIDE-MIP has the potential to rapidly classify RNA and DNA viruses in the clinic by bond-selective imaging of T and U residues.

### Identification of protein secondary structure in a single virus

As the profile in the amide I band is very sensitive to the protein secondary structures[53], WIDE-MIP can be a promising tool to characterize the protein structures compared to expensive and time-consuming approaches such as electron microscopy[54,55]. To explore such potential, we acquired the WIDE-MIP spectra of another DNA virus, VZV, shown in Fig. 5. It is reported that there are three envelope proteins, glycoprotein B, glycoprotein H, and glycoprotein L serving as the most essential VZV proteins that function as the core fusion complex[56]. These proteins have known 3D structure and all of them have a big proportion of β-sheet, and the proportion of turn cannot be ignored[57]. Figure 5a–c shows the defocused interferometric scattering image and bond-selective MIP image of amide II and amide I vibrations of single VZV viruses. Although there are some aggregates in the interferometric image, a lot of single virions are shown. Zoomed-in views of four single VZV viruses are illustrated in Fig. 5d–i. WIDE-MIP spectra of

these four single VZV (red arrows labeled in Fig. 5d–i) were further obtained (Fig. 5j). The specific IR peaks of DNA virus were observed at 1580 cm$^{-1}$ and 1612 cm$^{-1}$, indicating the vibrations of A, C and T residues in the viral DNA of VZV. Compared to the spectra of VSV and VACV, two broad peaks are observed at around 1630–1640 cm$^{-1}$ and 1668 cm$^{-1}$, which are assigned to the β-sheet and the turn structure in the viral proteins of VZV, respectively[53]. The statistical spectra acquired from 30 VZV virions (Fig. 5k) further reveal enriched β-sheet protein components, viral DNA, and lipids in VZV[58] (details in Supplementary Note 6). The spectral fidelity was confirmed by FTIR absorption spectrum of VZV powder (Fig. 5l). Therefore, besides the detection of major chemical components, WIDE-MIP can identify the protein secondary structure related to their function in a virus.

## Discussion

We present a single virus fingerprinting approach, termed WIDE-MIP microscopy, that addresses the unmet need for the identification of single virus. Our method allows composition detection of viral nucleic acids and proteins with high throughput. A theoretical framework for interferometric defocus-enhanced photothermal signal is developed and experimentally validated, providing a guideline to obtain a well-defined photothermal signal by adjusting the defocusing. Compared to scanning MIP, WIDE-MIP increases the imaging throughput by three orders of magnitude for fingerprint analysis of nanoparticles at the same SNR. Besides content detection of viral proteins, viral DNA, and viral RNA, WIDE-MIP further identifies the protein secondary structure in a single virus by revealing enriched β sheet components.

While we utilized defocused interferometric imaging to improve MIP contrast, our analysis of biological samples relies on MIP contrasts, whereas the interferometric images are solely used for sample localization before acquiring the MIP images. Despite the slight defocusing in the interferometric images, the MIP images remain in focus, revealing distinct contrasts of the biological samples. The MIP images not only provide valuable morphological information about the biological samples but also offer insightful biochemical details about their specific contents.

In this work, both VACV and VSV viruses were expressed with an eGFP envelope. The eGFP was fused to the VSV G protein, where each VSV contains -1200 molecules of the G protein on the viral surface[59]. With the formed G protein and G-eGFP fusion protein heterodimers, there are -600 eGFP molecules on the surface of a single virus. Comparing the size of eGFP ($2.4 \times 4.2$ nm) to that of the VSV virus ($80 \times 180$ nm), we estimate that only 1.3% of a single VSV virus consists

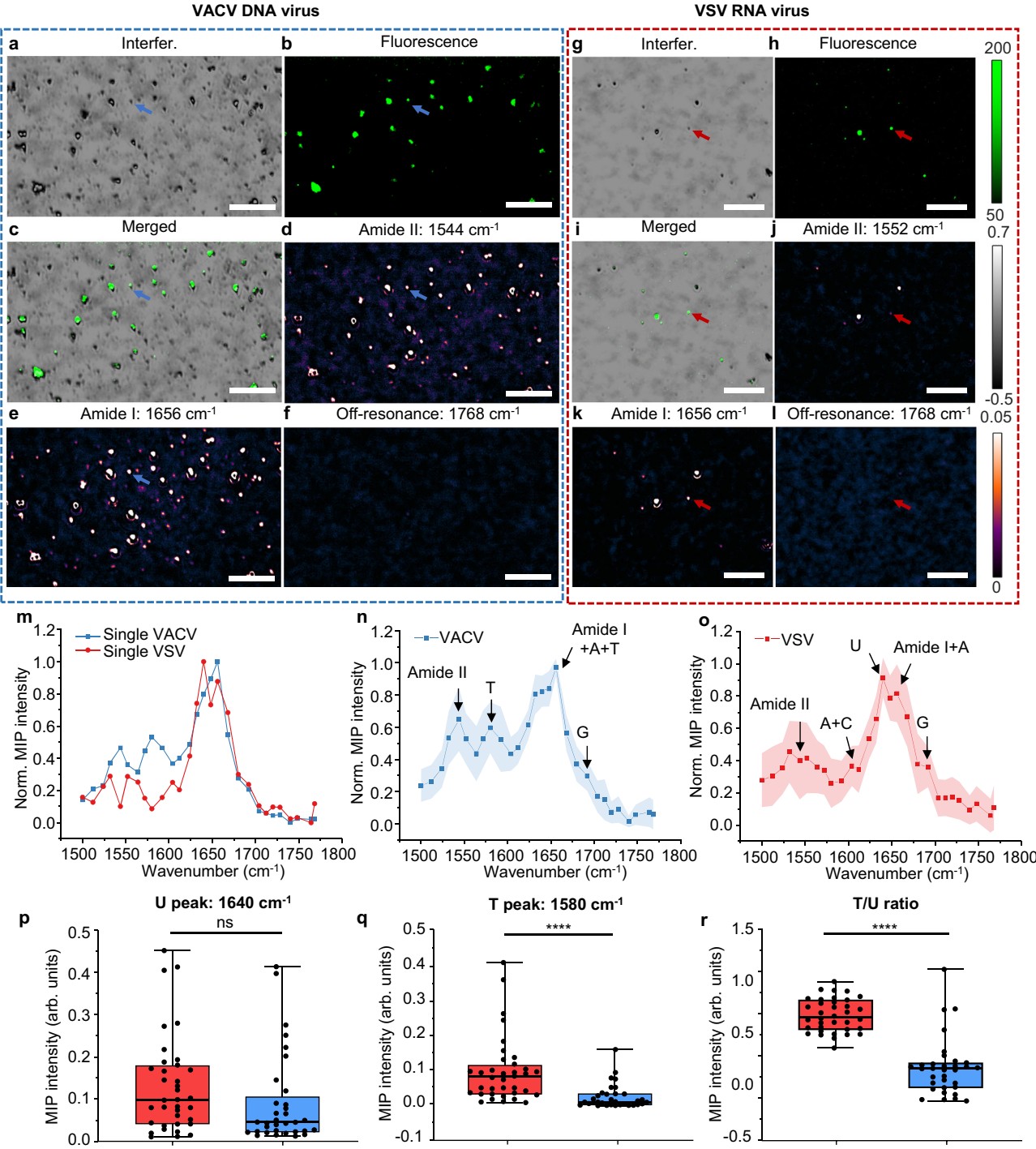

**Fig. 4 | Fingerprinting detection of single VACV and VSV viruses. a** Defocused interferometric scattering, **b** fluorescence, and **c** merged images of single VACV viruses. **d** Amide II bond-selective image of the same area with the pump at 1544 cm⁻¹. **e** Amide I bond-selective image of the same area with the pump at 1656 cm⁻¹. **f** Off-resonance image showed no contrast. **g** Defocused interferometric scattering, **h** fluorescence, and **i** merged images of single VSV viruses. **j** Amide II bond-selective image of the same area with the pump at 1552 cm⁻¹. **k** Amide I bond-selective image of the same area with the pump at 1656 cm⁻¹. **l** Off-resonance image showed no contrast. Scale bars: 10 µm. **m** MIP spectra of two single VACV and VSV viruses (blue and red arrows labeled). Statistical MIP spectra obtained from **n** 36 single VACV and **o** 33 VSV viruses. Error bands represent standard deviation of the mean. Power before the objective: pump: 22.9 mW at 1544 cm⁻¹, 29.1 mW at

1552 cm⁻¹, 34.5 mW at 1656 cm⁻¹, 35.8 mW at 1768 cm⁻¹, probe: -1 mW. Image acquisition time: 2.36 s per wavenumber. The MIP spectrum is normalized by the IR power. Quantified MIP contrast of peaks at **p** T residue ($P = 0.138892$) and **q** U residue ($P = 7.4 \times 10^{-5}$) of VACV and VSV. **r** Quantified MIP contrast ratio of peaks at T residue/U residue ($P = 5.2 \times 10^{-14}$) of VACV and VSV. $n = 36$ for biologically independent VACV samples and $n = 33$ for biologically independent VSV samples in **p**–**r**. The bound of box indicates 25% to 75% of data; inner line indicates medium; whiskers indicate maxima and minima of data. All statistical significance was analyzed using two-sided Student's *t* test. ns ($P \geq 0.05$) denotes no statistically significant difference. Asterisks **** ($P < 0.0001$) denotes statistically significant difference. Source data are provided as a Source Data file.

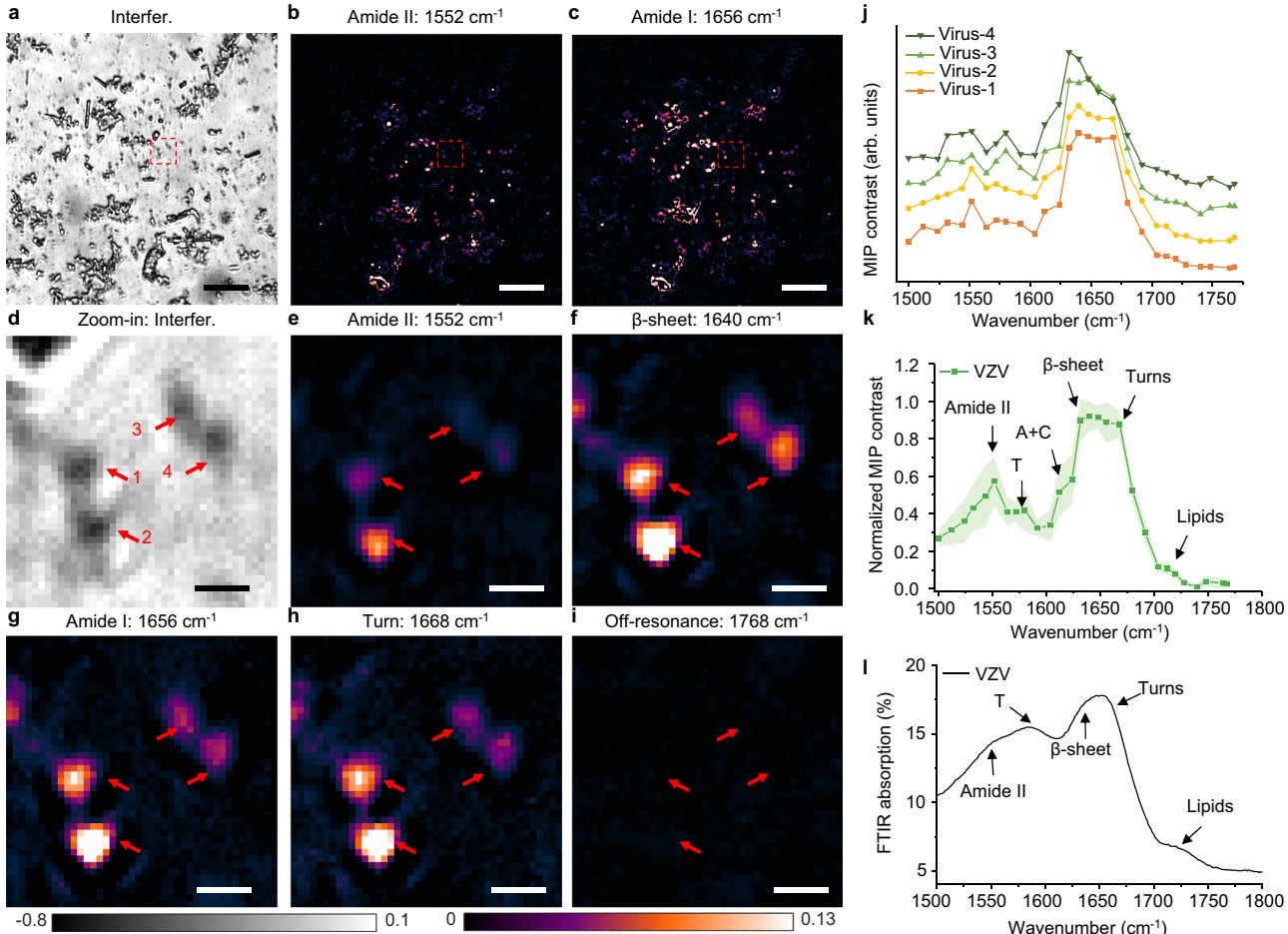

**Fig. 5 | Protein secondary structure identification in single VZV viruses.**
**a** Defocused interferometric scattering, **b** Amide II, and **c** amide I bond-selective image of single VZV viruses with the pump at 1552 cm⁻¹ and 1656 cm⁻¹, respectively. Scale bars: 10 μm. **d** Zoomed-in view of defocused interferometric scattering of four viruses in **a**. **e** Amide II, **f** β-sheet, **g** Amide I, **h** Turn bond-selective image of the same area in **d** with the pump at 1552 cm⁻¹, 1640 cm⁻¹, 1656 cm⁻¹, 1668 cm⁻¹, respectively. **i** Off-resonance image showed no contrast of the same area in **d** with the pump at 1768 cm⁻¹. Scale bars: 1 μm. **j** MIP spectra of four VZV viruses in **d** (red arrows labeled). **k** Statistical MIP spectra obtained from 30 single VZV viruses. Error bands represent the standard deviation of the mean. Power before the objective: pump: 29.1 mW at 1552 cm⁻¹, 33.1 mW at 1640 cm⁻¹, 34.5 mW at 1656 cm⁻¹, 34.1 mW at 1668 cm⁻¹, 35.8 mW at 1768 cm⁻¹, probe: ~1 mW. Image acquisition time: 2.36 s per wavenumber. The MIP spectrum is normalized by the IR power. **l** FTIR spectrum of pure VZV virus powder. The FTIR spectrum was acquired by an attenuated total reflection FTIR spectrometer. Source data are provided as a Source Data file.

of eGFP. Thus, the effect of eGFP on MIP imaging should be negligible due to the relatively low content of eGFP in a single virus.

Supporting this notion, eGFP has a β-enriched sheet structure, while no obvious β sheet chemical signature was observed in either VACV or VSV expressing eGFP in Fig. 4. Additionally, the pure VZV viruses without any labeling showed an enriched β sheet component. These findings further support that the effect of eGFP on MIP imaging is minimal. Moreover, we note that the key distinction between DNA and RNA viruses lies in the different nucleic acid peaks associated with T and U residue vibrations, which are also unrelated to the eGFP proteins.

For the analysis of actual virus samples, label-free methods may be more suitable for diagnostic purposes. Thus, in this work, we first performed fluorescence-guided MIP analysis of single viruses by integrating fluorescence imaging and MIP imaging for accurate virus identification via WIDE-MIP (Fig. 4). Subsequently, we performed MIP imaging and obtained fingerprint spectra of unlabeled pure VZV viruses to achieve label-free detection of single viruses (Fig. 5). These approaches allow for comprehensive analysis while minimizing any potential influence of fluorescence labeling on the MIP imaging results.

In comparison to recently reported fluorescence-detected MIP (F-MIP) microscopy[60,61], WIDE-MIP offers a distinct advantage in detecting

bionanoparticles with low levels of expressed fluorescence tags (Supplementary Fig. 9, details in Supplementary Note 7). Although the photobleaching of aggregated eGFP-VACVs showed a similar level in F-MIP[60] (<~10%), severe photobleaching was observed in single VACVs (>~95%). This photobleaching of single viruses limits the detection of photothermal modulation and acquisition speed. Considering that the MIP signal relies on the difference in fluorescence intensity between the IR-on (hot) and IR-off (cold) states, this severe bleaching at the single-virus level further compromises the reliability of F-MIP. Instead, we focused on fluorescence-guided WIDE-MIP analysis, which enables label-free chemical imaging of single viruses.

Benefit from the compositional analysis of single viruses in a label-free manner, we envision WIDE-MIP as an alternative analysis tool for viral vectors used in gene therapy. Viral vectors, including adeno-associated viruses, adenoviruses, and lentiviruses, are increasingly used in gene therapy but pose challenges for quality control testing and characterization due to their complexity[62,63]. To ensure a safe, consistent, and high-quality product, accurate and rapid analytical assays are needed to monitor quality attributes. Sodium dodecyl-sulfate polyacrylamide gel electrophoresis, mass spectrometry, immunoblotting, enzyme-linked immunosorbent assay, polymerase chain reaction, or transmission electron microscopy are used to

identify protein, genome, and capsid content[64,65], but these assays can be time-consuming and require pre-treatments or extraction. To address these limitations, we further demonstrated high-speed chemical imaging of single VACV by reducing the acquisition time to 0.32 s per image per wavenumber of single viruses and the SNR of one single VACV is ~ 4 within the field of view of 24 by 24 μm (Supplementary Fig. 10). With the ability to rapidly acquire fingerprints of single viruses, WIDE-MIP can provide insights into the quality control of viral vectors, such as their identity, purity, and stability[64] (details in Supplementary Note 8).

Future improvement for WIDE-MIP can focus on fingerprinting viruses or exosomes in liquid conditions, allowing for detecting biological nanoparticle samples in their natural states. This can be achieved by incorporating microfluidic systems[18] and designed substrates[43] to capture viruses and enhance imaging contrast in liquid measurements, which will broaden the applicability of WIDE-MIP to real-world applications.

## Methods

### Materials
The double-side polished silicon wafer (4 inch, 500 μm thickness) was purchased from University Wafer and diced into 10 mm × 20 mm pieces. PMMA nanoparticles were purchased from Phosphorex. 0.1% poly-l-lysine and bovine serum albumin (BSA) were purchased from Sigma-Aldrich. Inactivated VZV strain VZ-10 was purchased from Fisher Scientific.

### Lab-built WIDE-MIP microscope
The IR pump beam was generated by a tunable (from 1400 to 1800 cm$^{-1}$) mid-IR laser (Firefly-LW, M Squared Lasers) operating at 20 kHz repetition rate with a ~20 ns pulse duration. The pump pulses were modulated by an optical chopper (MC2000B, Thorlabs). For widefield photothermal imaging, the IR beam was weakly focused on the sample plane from the bottom of the silicon substrate via an off-axis parabolic mirror. A delay pulse generator (9254, Quantum Composers) was used to synchronize the pump pulse, the probe pulse and the interferometric pulse recorded by the camera. For the power normalization, a power meter (PM16-401, Thorlabs) was used to monitor the IR power. The visible probe was provided with a pulsed 520 nm nanosecond laser (NPL52C, Thorlabs) with a pulse duration of 129 ns. The probe laser illuminated the sample from the top through a 50/50 polarizing beam splitter, a quarter-wave plate and a high numerical aperture (NA) air objective (MPLFLN Olympus, ×100, NA 0.9). To acquire the defocus-enhanced photothermal images, the objective was adapted with an objective piezo scanner (Piezosystemjena, MIPOS 100), which can provide precise Z axis scanning in steps of 100 nm. The incident light was then scattered by the sample and reflected by the silicon substrate. The consequent interferometric scattering was collected by the same objective and recorded by a complementary metal-oxide semiconductor (CMOS) camera (Q-2HFW, Adimec). We further employed a 2 million well-depth camera to receive sufficient probe photons at each pixel.

### Theoretical simulation
A theoretical framework was developed to calculate the focus-dependent interferometric and photothermal images of nanoparticles of different sizes. An image field representation of optical fields was employed by considering imaging optics and system parameters. The simulation was built upon the previously developed model via the BEM, which is a computationally efficient approach for calculating the interferometric scattering from a nanoparticle near a substrate. A custom-developed MNPBEM toolbox was used to solve Maxwell's equations for a dielectric environment where the nanoparticles have homogeneous and isotropic dielectric functions and are separated by abrupt medium interfaces[43]. The MNPBEM was implemented in MATLAB and the simulation could be compartmentalized into five steps: (1) dielectric functions initialization of nanoparticle, substrate and environment to define the system geometry, such as the radius ($r$) and refractive index ($n$) of the nanoparticle; (2) specification of excitation scheme, such as incident illumination wavelength ($\lambda$) and illumination function; (3) solver setup for the BEM equations; (4) BEM equations' solutions for the given excitation; (5) calculation of the far-field scattered field and image fields of the nanoparticle. We assumed that a PMMA particle ($r$, $n = 1.49$[66]) was placed on top of a silicon substrate ($n = 4.2$[67]). The interferometric scattered field was calculated as the total backscattered field considering the reflections from the silicon surface using Green's functions. The image fields were then simulated via angular spectrum representation integral and the detected interferometric signals were calculated according to Eq. (2) in the main text. The photothermal signals were then generated from the interferometric scattering difference between IR-on (hot) and IR off (cold) states. We simulated the interferometric images of a PMMA bead with different sizes at both cold ($T = 293.15$ K) and hot ($T = 373.15$ K) states along the Z axis scanning of objective. The interferometric and photothermal contrasts were recorded at the center of the diffraction-limited image of the PMMA bead as the focus position Z sweeping. Here, Z was set to be zero for exact optical focusing at the sample-substrate interface for the light-collecting objective, where the numerical aperture of the objective was also considered for the collecting angle. Thus, the defocus curves of both interferometric and photothermal contrasts were obtained.

### Fluorescence imaging of single virus
A 488 nm diode laser (200 mW, Cobolt 06-MLD) was used for fluorescence excitation. The excitation beam was expanded through a 4f system ($f_1 = 50$ mm, LA1131-A-ML and $f_2 = 300$ mm, LA1484-A-ML, Thorlabs) and then coupled into the light path of probe laser. The fluorescence emission went through the same objective lens and was collected with a filter set (Excitation filter: FES0500, Thorlabs; Dichroic beam splitter: Di03-R405/488/532/635-t1-25×36, Sermock; Emission filter: FF01-525/30-25, Sermock). A CMOS camera (FLIR, Grasshopper3GS3-U3-51S5M) was used to capture the fluorescence images and the exposure time was set to 5 s for optimized contrast. Virus samples on silicon substrate were first imaged by fluorescence to confirm the single virus and then imaged with WIDE-MIP at the same position.

### Data processing
The interferometric and MIP images were acquired using a lab-built Labview program and analyzed with ImageJ, detailed methods were described in previous work[39]. The interferometric images were captured at a camera shutter speed of 1270 Hz. The MIP images were obtained as the intensity difference between the hot and the sequential cold frame, at the speed of 635 frames/s. The interferometric images were normalized by the background reflection. Pseudocolor was added to the MIP and fluorescent images with ImageJ software or MATLAB. The SNR was calculated from the ratio between the mean value of the center region (25 pixels) of the single particle in resonance photothermal image and the standard deviation of the off-resonance photothermal.

### Sample preparation
The silicon wafers were cleaned in sequence with acetone, ethanol, and deionized (DI) water rinse. For PMMA nanoparticle detection, the PMMA beads were diluted ~100 times with DI water and then spin-coated on the silicon substrate and dried in air. For virus analysis, the VACV and VSV samples were prepared according to the previous method[38]. Both the recombinant VSV expressed an eGFP and VACV expressed Venus. To load viruses onto the substrate, the silicon

surface was incubated with 0.1% poly-l-lysine for 1 h. Then, 100 µL of either VACV or VSV stock was incubated in the center of each poly-L-lysine coated silicon for 1 h at room temperature. Both VACVs and VSVs were diluted to ~$1 \times 10^8$ PFU/mL. All virions were crosslinked and inactivated using 1.0 mL of 4% formaldehyde for 1 h. After modification, the substrate was rinsed with sterile filtered DI water and then dried in air. For VZV detection, the lyophilized VZV pellet was dissolved in 200 µL PBS and filtered by a 0.22 µm filter. Then, 100 µL of VZV stock was incubated in the center of each poly-L-lysine coated silicon for 1 h at room temperature. After modification, the substrate was rinsed with sterile filtered DI water and then dried in air. To fabricate the pure protein film sample, 10 µL 10 mg/mL BSA solution was dropped onto the silicon surface and dried in air. The pure DNA and RNA solutions were prepared from the cDNA of melanoma cells and ssRNA of T24 cell, respectively, as described earlier[68]. The pure DNA and RNA films were prepared by dropping 5 µL cDNA and ssRNA onto the silicon surface, respectively, and dried in air.

### Cell Lines
Melanoma cell line (1205Lu) was obtained from Dr. Meenard Herlyn (The Wistar Institute). T24 cell line (Cat#: HTB-4) was purchased from the American Type Culture Collection (ATCC). All cell lines were authenticated and tested to be mycoplasma negative.

### FTIR measurement
The FTIR spectra of 200-nm dry PMMA beads and viruses were measured on an attenuated total reflection FTIR spectrometer (Nicolet Nexus 670, Thermo Fisher Scientific). The spectra resolution is 2 cm$^{-1}$ and each spectrum was measured with 128 scanning. All spectra were automatically normalized by the baseline correction on the system.

### Statistics and reproducibility
All experiments were independently repeated at least three times with similar results. The sample sizes for all statistical experiments exceeded 10. No statistical method was employed to predetermine the sample size. All data collected during the experiments were included, no data were excluded from the analyses.

### Reporting summary
Further information on research design is available in the Nature Portfolio Reporting Summary linked to this article.

## Data availability
All the imaging raw data related to this work are available in Zenodo (https://doi.org/10.5281/zenodo.7957859). Source data are provided with this paper.

## Code availability
MATLAB codes for simulation in this study are available in Zenodo (https://doi.org/10.5281/zenodo.7957859).

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

## Acknowledgements

This research was supported by NIH grant R35GM136223 to J.-X.C. The authors thank Fukai Chen for providing the pure DNA and RNA samples.

## Author contributions

Q.X. and J.-X.C. designed the experiments. Q.X. performed the experiments and analyzed the data. Z.Y.G., H.N.Z. and C.Y. helped in data analysis. Z.Y.G. and Q.X. constructed the setup. Z.Y.G. coded the program for data acquisition. C.Y., H.N.Z. and M.S.U. provided guidance and discussions on the simulation. S.S. prepared the VACV and VSV samples and provided constructive suggestions for virus analysis. L.W. performed the atomic force microscope analysis. Q.X. wrote the manuscript with contributions from J.-X.C., Z.Y.G., C.Y. and H.N.Z. J.-X.C. and J.H.C. provided overall guidance for the project. All authors have given approval to the final version of the manuscript.

## Competing interests

The authors declare no competing interests.
