## [Peer Review File · Nature Communications]

REVIEWERS' COMMENTS

Reviewer #1 (Remarks to the Author)

In this work, authors present the development of a widefield interferometric defocus-enhanced MIP microscope (WIDE-MIP), building upon conventional MIP to enhance the contrast, achieve small analytical spots and reduce the acquisition time to sub-second per imager per wavenumber. The work is important for the composition determination in biological materials with small sizes, such as viruses. The authors differentiated DNA and RNA through specific residues' vibrational modes and monitored capsids' secondary structure. Due to the important evolution comprising novel instrumentation, the capability of analysing single viruses' hyperspectral images/spectra, well-supported claims, and promising applications, I recommend the work for publication. Please check the comments below for suggestions.

1. The abstract would benefit from a revision to improve the precision; "current methods" – which ones? "low-abundance chemical contents" – how low? Compared to what?
2. In the simulated interferometric images of the PMMA bead, why is the difference between hot and cold states exactly 1 K?
3. The spectral variation in Figure 3c is more pronounced in the 1500-1575 cm^{-1} range compared to the rest of the spectrum. This was, however, the region used to acquire most of the data regarding viral particles. Should readers expect an inherently larger error in this region for all samples?
4. Please comment on the maximum theoretical spatial resolution of the technique.
5. How is the technique affected by a higher water content in other biologic structures such as whole cells? Will proteins' secondary structure contents estimation be affected?

Reviewer #2 (Remarks to the Author)

In this work, the authors demonstrated single virus fingerprinting by means of mid-infrared photothermal (MIP) microscopy based on widefield interferometric scheme. They characterized the relationship between the focus positions and MIP and interferometric image contrast to optimize MIP contrast, and they successfully improved contrast of interferometric MIP imaging by defocusing. The authors also demonstrated fingerprinting of single tiny viruses through hyperspectral MIP imaging with a 3 order of magnitude higher throughput. Biochemical components in various viruses, such viral proteins and nucleic acid and secondary structures of viral proteins in a single virus were identified by the widefield interferometric defocus-enhanced MIP imaging.

MIP microscopy is a state-of-the-art super-resolution IR imaging leveraged in various scientific fields and the authors have been dedicated to developing novel types of MIP microscopy and improving its performance. In interferometric MIP imaging, as the authors rightly pointed out, optimal contrast in MIP imaging has not yet been obtained, hence improving MIP contrast by characterizing focus positions is highly useful for the corresponding fields. However, although I understood the importance of the technical advancement of MIP microscopy mentioned above, it is minor and not as sophisticated as previous studies by the authors (Zhang et al., *Sci Adv*, Yin et al., *Nat. Commun.*, Bai et al., *Sci Adv.*, Zhao et al., *Nat Commun.*). The present work seems to be more incremental than novel to be published in *Nature Communications* that requires notable conceptualization and technical advancement. It also seems that the technique demonstrated here can be only applied to samples with simple structures, such as beads and viruses, and cannot be employed in a broad range of samples that have complex structures, such as cells. In addition, the claims and validation of single virus identification are not entirely supported by the results in this paper.

Thus, the manuscript describes the powerful methodology that is highly valuable for researchers working with MIP microscopy, and should be published in other high-impact journals, but I could not recommend the manuscript to be published in *Nature Communications*.

Here's my comments on the manuscript that can make the manuscript significantly better.

1. In Fig3 a,b, slightly bright signal showing beads-like structures in the background of the image can be seen. The authors should explain this.
2. Fig 4 shows comparison of fluorescence and MIP images. However, since the viruses are stained by GFP, I wonder the effect of GFP on MIP imaging. The discussion or explanation of the effect should be provided.
3. Since the spectral resolution of MIP spectra is not sufficient to differentiate a few IR peaks overlapped with each other, fingerprinting viruses is ambiguous. The authors utilize QCL as an IR light source that could show the variation and the fluctuation of IR intensity. The intensity fluctuation might not be accurately compensated especially the effect could be dominated when the data point spacing is large. The authors should clarify that such low spectral resolution could identify the specific IR molecular vibrations in the spectra.
4. Related to Comment 3, complex IR spectra from biological samples are usually decomposed to accurately characterize vibrational modes of their biochemical components. Unexpected changes of spectral features, such as peak splitting of the amide I band and shift of peak positions due to interactions of molecules and molecular states, were frequently seen in IR vibrational analysis of biological samples. Especially, IR spectra that the authors showed has the much lower spectral resolution than that in other literatures, which may result in misinterpretation of vibrational signatures in such complex IR spectra. The authors should state validness of their results on fingerprinting single viruses.
5. The authors claimed that vibrational peaks originating from amino acid residues could be seen in MIP spectra of viruses. However, I am not sure that the amino acid components were dominantly reflected on IR spectra of virus samples, in which many other compartments coexist, such as nucleocapsid proteins and envelopes (lipids, proteins). The authors should justify their MIP spectra exhibiting DNA/RNA compartments by discussing the number of biochemical components in viruses. In addition, FTIR spectra of the virus samples also verify the successful demonstration of fingerprinting of viruses.
6. The authors described the future of rapid diagnostic tools using WIDE-MIP that they demonstrated. This is nice view and highly demanded for the current world, however, it is difficult to see that this method could work for clinical samples of saliva or breath condensates, which have a lot of foreign substances without washing or purifying processes. Usually, such non-target substances in clinical samples either deteriorate the detection sensitivity of techniques or make it difficult to probe specific vibrational modes of target molecules. It is great perspective, but I feel it seems to be exaggerating a little bit.

Minor comments

7. In the introduction, the authors cited several papers on MIP microscopy, but most of them are from the same group. As far as I have recognized this fields, there are a few more groups that develop MIP microscopy for biological samples (Samolis et al., *Biomed. Opt. Express*, (2021)/ Kato et al., *Anal. Sci.*, (2022)/ Lim et al., *J. Phys. Chem. Lett.*, (2019)). Hence, it would be better to cite those literatures to show readers that MIP microscopy has been recently emerged and attracted many attentions from a broad spectrum of researchers.
8. In this work, viruses were not measured in liquid condition, but ambient condition. This might give influence on physical and biochemical properties of viruses. Could the authors give comments on that?
9. In previous work from the authors, instead of scattering-based MIP imaging, MIP imaging based on temperature-modulated fluorescence intensity was demonstrated and they improved the sensitivity by two orders of magnitude. The authors may want to add discussion about the implementation of the technique or comments on that.

Reviewer #3 (Remarks to the Author)

In this research article, Qing Xia et al. propose a novel microscopy method for non-destructive imaging, evaluation and characterization of viruses. Proposed technique involves immobilization of viruses on a silicon substrate and illumination with visible laser beam. When particles are illuminated with mid-infrared laser, the infrared absorption characteristics of the particle changes the scattering properties which modulates the scattered and detected visible signal.

The work presented in this article is certainly novel. Authors have done a fair job capturing the previous work on this field and have successfully identified the original aspects of their work. Wide-field imaging of single viral particles and analysis of their photo-thermal spectrum individually is challenging because of the small size of these particles. However, authors argue that combination of widefield interferometric imaging enables setting up the detection plane to an optimum z-location so that MIP sensitivity is enhanced.

Very interestingly, authors demonstrate spectral response to tunable mid-infrared illumination can be used to fingerprint the individual viruses by characterizing the biochemical contents. In one example, they can distinguish if the imaged viruses have RNA or DNA in them by identifying Uracil and Thymine signatures in the detected MIP spectrum. In another example, authors demonstrate this technique can be used to identify the protein secondary structures in the viruses.

However, the paper can be improved in several ways.

I found it very confusing how the technique is explained in the methods section. It looks like all of Fig.2 is dedicated to convincing the reader that "interference contrast and MIP contrast peaks at different planes", but not clear how this information is later used. Where the MIP contrast peak occurs depends on the particle size, it would be set at a certain Z-height, and the photo-thermal data would be collected at that focal plane for particle characterization. So, in essence, one can do a Z-scan of MIP contrast, and leave it at maximum contrast point. It is confusing how the interferometric contrast and its dependence to Z-scan is used.

Another confusing part is what the images actually represent. When we are looking at the photo-thermal images in the paper (i.e. Fig-1b, Fig-3a and 3b, etc), are these what is seen on the CMOS camera? Since the MIP signal is the modulation of visible signal by the photo-thermal effect, I assume these images represent how the visible reflectance/scattering pattern changes, so maybe there was some subtraction of original (Cold) state.

Maybe readers who are very familiar with the method will not encounter the similar confusion, but unless that's the case, the readers will likely not understand the method well.

One final recommendation for the methods section - the depth of focus for the imaging system needs to be provided, since results has a Z-dependence.

I think the paper has solid scientific basis, but conclusions towards utilizing this tool for diagnostic applications may not be well suited. First of all, even though it is called to be a wide-field imaging tool, the field of view is maybe about 50um wide. If the goal is to analyze a clinical sample, what would be the mass transport rate needed to immobilize detectable number of viruses on the surface and within a small square like this. Even if the mass transport issue is resolved, it's not clear how the viruses are identified within a noisy background as in Figures 4a and 4b. For the data in Fig. 4, authors used GFP to identify the viruses from the background, which is of course not possible in a label-free diagnostic situation. So, how would they be identified from the background? Is the MIP spectrum good enough to call there is a virus in any particular location or sample?

Diagnostic applications require simple architectures that can be robustly implemented in very different settings.

On the other hand, research and development tools has the flexibility to be complex and costly.

Considering development of viral vectors for a vast number of different therapeutic applications gained popularity in the recent year, I recommend the authors to look into applications such as analysis and quality control of viral vectors. The proposed technology is certainly novel and has strengths, but it may be more suitable for applications such as this one.

RESPONSE TO REVIEWERS' COMMENTS

Reviewer #1

General comments: In this work, authors present the development of a widefield interferometric defocus-enhanced MIP microscope (WIDE-MIP), building upon conventional MIP to enhance the contrast, achieve small analytical spots and reduce the acquisition time to sub-second per imager per wavenumber. The work is important for the composition determination in biological materials with small sizes, such as viruses. The authors differentiated DNA and RNA through specific residues' vibrational modes and monitored capsids' secondary structure. Due to the important evolution comprising novel instrumentation, the capability of analysing single viruses' hyperspectral images/spectra, well-supported claims, and promising applications, I recommend the work for publication. Please check the comments below for suggestions.

Author reply: We thank the referee for reviewing our manuscript, affirming its clarity and comprehensibility, and for her/his suggestions on how to improve the manuscript. The provided comments and questions are responded point by point below.

Comment 1: The abstract would benefit from a revision to improve the precision; “current methods” – which ones? “low-abundance chemical contents” – how low? Compared to what?

Author reply: We thank the reviewer's suggestions. Diagnostic tests on the viruses commonly rely on the detection of nucleic acids or surface proteins. The abundance of viral protein and viral nucleic acid in a single virion varies widely depending on the viral structure and its genome size. For a vesicular stomatitis virus, the total number of proteins is about 5000 (J. Virol., 1985, 54, 598-607), which is much lower than the abundance of proteins in a bacterium (~ 1 million). We have added this information to the **Abstract** as follows.

“Clinical identification and fundamental study of viruses rely on the detection of viral proteins or viral nucleic acids. Yet, amplification-based and antigen-based methods are not able to provide precise compositional information of individual virions due to small particle size and low-abundance chemical contents (e.g., ~ 5000 proteins in a vesicular stomatitis virus).”

Comment 2: In the simulated interferometric images of the PMMA bead, why is the difference between hot and cold states exactly 1 K?

Author reply: We appreciate the comment. Previously, we set $\Delta T = 1$ K just as an example and we mainly focused on the analysis of the focal plane difference between interferometric and MIP images.

To provide a rigorous result, we further did the COMSOL simulation to obtain the accurate temperature difference between hot and cold states. The calculated transient temperature rises ΔT for a single 200 nm PMMA bead after vibrational excitation by a single IR pulse is found to be ~80 K over a temporal window of 129 nanoseconds (revised **Figure S2** or see **Supplementary note 1** below).

“Supplementary Note 1. Simulation of the temperature rise.

To better understand the photothermal process, a theoretical model was built to solve the temperature difference between hot and cold states. This model was developed using COMSOL based on our previous work¹. The time-dependent thermal diffusion process can be simulated via the heat-transfer-

in-solids module in COMSOL Multiphysics². To calculate the heat dissipation, a heat source term $Q(t)$ is defined as below:

$$C_p \rho \frac{\partial T}{\partial t} + \nabla \cdot (-k \nabla T) = Q(t) \quad (1)$$

where T is the temperature, t is the time, C_p is the heat capacity, ρ is the density, and k is the thermal conductivity of the material in the system.

In this simulation, a 200 nm PMMA bead was sitting on the silicon substrate in air. The IR heating beam size is 24.6 μm by 30 μm and the power is 40 mW measured from the experiments. Both the initial temperature and the simulation boundary were assumed to be 298 K. The heat source was defined as the domain of the PMMA bead. Heat convection was not considered in this simulation. By solving the equation (1), the temperature rise distribution of the bead under single IR pulse heating was simulated via COMSOL 6.0. Supplementary Fig. 2a shows the temperature profile of the system. The calculated ΔT on the single bead is ~ 80 K, integrated from the pulse width of single probe pulse, which is ~ 129 ns (Supplementary Fig. 2b)."

Supplementary Fig. 2 Simulated temperature rise of a 200 nm PMMA bead under single IR pulse heating. (a) Temperature distribution of a 200 nm PMMA bead on the silicon substrate heated by a single IR pulse. Time is at 400 ns after the rising edge of the IR pulse. (b) Simulation results of thermodynamic properties of the heated PMMA bead.

We then set $\Delta T = 80$ K and redid the BEM simulation. For the experiment results, the previous MIP contrast was calculated from the subtraction of the hot and cold raw image intensities for the sake of simplicity. In order to provide a more accurate comparison between the experimental results and the simulation value, we recalculated the MIP contrast in the manuscript (revised **Figure 2g and i** or see below): the MIP image results were calculated from the subtraction of the hot and cold raw image intensities and further divided by the reflection intensity of the substrate to obtain the MIP contrast. Since the MIP image is acquired at the single wavenumber 1728 cm^{-1} , the normalization of the IR power is not needed when compared to the simulation results. It should be pointed out that this revision does not affect all the results of the original MIP image and spectra in the manuscript, just to provide a more accurate comparison with the theoretical MIP contrast value. After revision, the simulated MIP contrast value matches the experimental MIP contrast very well when ΔT is set as 80 K. Thus, the the temperature difference between hot and cold states should be ~ 80 K.

We thank the reviewer for bringing this to our attention. We have added new **Supplementary Note** in supporting materials and revised the **Fig. 2** based on $\Delta T = 80$ K as follows.

Fig. 2 Simulation and experimental validation of interferometric defocus-enhanced photothermal contrast. PMMA beads of $D = 200$ nm were used as the testbed. (a) Simulated defocus curves of interferometric contrast at the cold ($T = 293.15$ K) and hot ($T = 373.15$ K) states. Zoomed-in view of simulated defocus curves of interferometric contrast at the position of (b) $Z_1 = 0$ μm and (c) $Z_2 = 0.4$ μm . Interfer.: Interferometric. The photothermal contrast is 0.6% at $Z_1 = 0$ μm and 0.00035% at $Z_2 = 0.4$ μm . (d) Simulated interferometric images at $Z_2 = 0.4$ μm , $Z_1 = 0$ μm (left) and interferometric scattering along Z axis (right). (e) Simulated MIP images at $Z_2 = 0.4$ μm , $Z_1 = 0$ μm (left) and MIP imaging along Z axis (right). (f) Experimental interferometric images at $Z_2 = 0.4$ μm , $Z_1 = 0$ μm (left) and interferometric scattering along Z axis (right). (g) Experimental MIP images at $Z_2 = 0.4$ μm , $Z_1 = 0$ μm (left) and MIP imaging along Z axis (right). Scale bar: 500 nm. All Z axis images are obtained from $Z = -1$ to 1 μm . (h) Simulated and (i) experimental defocus curves of interferometric and MIP contrast. Power before the objective: pump: 48 mW at 1728 cm^{-1} , probe: ~ 1 mW. Image acquisition time: 2.36 s per image. Z-axis scanning step: 100 nm.

Comment 3: The spectral variation in Figure 3c is more pronounced in the 1500-1575 cm^{-1} range compared to the rest of the spectrum. This was, however, the region used to acquire most of the data regarding viral particles. Should readers expect an inherently larger error in this region for all samples?

Author reply: We thank the reviewer for raising the concern of spectral variation in 1500-1575 cm^{-1} range, which is the region used to acquire most of the data regarding viral particles. We apologize for any confusion caused by the original Figure 3. We have acknowledged and corrected the mistake

made in Figure 3, where the MIP contrast values at 1512, 1564, and 1612 cm^{-1} were mistakenly calculated as negative values.

To further estimate the stability of WIDE-MIP, we chose the off-resonance region of PMMA vibration from 1510 to 1610 cm^{-1} and calculated the standard deviation of the mean MIP contrast in this range. The result showed that the standard deviation of the mean MIP contrast was $\sim 0.16\%$, indicating the stable hyperspectral performance of WIDE-MIP in this spectral range. Therefore, readers should expect reliable data from this region for all samples. The revised discussion about the spectral noise and the revised Figure are shown as follows.

“The statistical spectra of 30 individual beads showed the distinguished resonance peaks of both C–H and C=O stretching vibrations (Fig. 3c, red line). The standard deviation of the mean MIP contrast within the range of ~ 1510 to 1610 cm^{-1} was found to be $\sim 0.16\%$, which corresponds to the off-resonance region of PMMA vibration. This demonstrates that the stable hyperspectral performance of WIDE-MIP. Furthermore, the spectral fidelity was confirmed by comparing the WIDE-MIP spectrum with FTIR absorption spectrum of PMMA (Fig. 3c, black line)⁴².”

Fig. 3 Hyperspectral performance and spectral fidelity of WIDE-MIP microscopy. MIP image of $D = 200 \text{ nm}$ PMMA beads with IR excitation at (a) 1452 cm^{-1} and (b) 1728 cm^{-1} . Scale bars: $5 \mu\text{m}$. (c) MIP spectrum (red) and FTIR spectrum (black) of $D = 200 \text{ nm}$ PMMA beads. The averaged MIP spectrum was obtained from 30 single PMMA beads. Power before the objective: pump: 31.4 mW at 1452 cm^{-1} , 38.6 mW at 1728 cm^{-1} , probe: $\sim 1 \text{ mW}$. Image acquisition time: 2.36 s per wavenumber. The MIP spectrum was normalized by the IR power. The FTIR spectrum was acquired by an attenuated total reflection FTIR spectrometer.”

Comment 4: Please comment on the maximum theoretical spatial resolution of the technique.

Author reply: Mid-infrared photothermal (MIP) microscopy commonly yields submicrometer spatial resolution. In a typical MIP microscope, a visible probe beam detects the photothermal chemical contrast induced by a vibrational absorption. When the wavelength of the pump mid-IR beam is tuned to the absorption peak of the sample, the thermal effect will lead to a local temperature change and consequently thermal expansion, pressure wave emission, change of refractive index, and change of Grüneisen parameter. By measuring the corresponding divergence variation in the copropagating probe beam, a photothermal chemical contrast can be acquired. Thus, the submicron resolution is reached according to the diffraction limit of the probe wavelength in the visible region.

Due to different properties of the sample and various setup of the imaging system, it is difficult to calculate the maximum theoretical spatial resolution in general. However, we can obtain higher spatial resolution by improving the experimental setup or detection method. Computational imaging-enhanced wide-field interferometric microscopy enabled interferometric scattering imaging of single biological nanoparticles with a resolution of $\sim 150 \text{ nm}$ (ACS Nano, 2020, 14, 2002-2013). It provides us the possibility to improve the spatial resolution of WIDE-MIP.

In addition, although the maximum theoretical resolution is limited by the diffraction limit of probe light,

Zhang et al developed a novel detection method for MIP imaging, which achieved super-resolution imaging of non-fluorescent molecules through high-harmonic demodulation in photothermal relaxation (Nat. Photon., 2023, 17, 330-337). The authors demonstrated label-free bond-selective electron and vibrational photothermal relaxation imaging of living cells at 280 nm and 120 nm resolution, respectively.

Comment 5: How is the technique affected by a higher water content in other biologic structures such as whole cells? Will proteins' secondary structure contents estimation be affected?

Author reply: The referee raised a good point. The protein amide I band is overlapped with the bending vibration of water around 1650 cm^{-1} . In this case, we could replace the water medium with D_2O phosphate buffered saline to avoid the resonant absorption. In a recent study, we have determined the protein secondary structure using this method (arXiv:2302.11769).

Reviewer #2

General comments: In this work, the authors demonstrated single virus fingerprinting by means of mid-infrared photothermal (MIP) microscopy based on widefield interferometric scheme. They characterized the relationship between the focus positions and MIP and interferometric image contrast to optimize MIP contrast, and they successfully improved contrast of interferometric MIP imaging by defocusing. The authors also demonstrated fingerprinting of single tiny viruses through hyperspectral MIP imaging with a 3 order of magnitude higher throughput. Biochemical components in various viruses, such viral proteins and nucleic acid and secondary structures of viral proteins in a single virus were identified by the widefield interferometric defocus-enhanced MIP imaging. MIP microscopy is a state-of-the-art super-resolution IR imaging leveraged in various scientific fields and the authors have been dedicated to developing novel types of MIP microscopy and improving its performance. In interferometric MIP imaging, as the authors rightly pointed out, optimal contrast in MIP imaging has not yet been obtained, hence improving MIP contrast by characterizing focus positions is highly useful for the corresponding fields. However, although I understood the importance of the technical advancement of MIP microscopy mentioned above, it is minor and not as sophisticated as previous studies by the authors (Zhang et al., Sci Adv, Yin et al., Nat. Commun., Bai et al., Sci Adv., Zhao et al., Nat Commun.). The present work seems to be more incremental than novel to be published in Nature Communications that requires notable conceptualization and technical advancement. It also seems that the technique demonstrated here can be only applied to samples with simple structures, such as beads and viruses, and cannot be employed in a broad range of samples that have complex structures, such as cells. In addition, the claims and validation of single virus identification are not entirely supported by the results in this paper. Thus, the manuscript describes the powerful methodology that is highly valuable for researchers working with MIP microscopy, and should be published in other high-impact journals, but I could not recommend the manuscript to be published in Nature Communications.

Author reply: We sincerely appreciate the valuable comments from the referee. We would like to address the concerns and further clarify the novelty and significance of our research.

We appreciate the high evaluation of our previous studies (Zhang et al., Sci Adv, Yin et al., Nat. Commun., Bai et al., Sci Adv., Zhao et al., Nat Commun.) by the reviewer. Meanwhile, we would like to emphasize that our current work introduces several innovations that enabled fingerprint analysis of single viruses, as summarized below.

- 1) Enhanced Sensitivity: Compared to our earlier work (Science Advances 2019) where a LED was used as a probe, we introduced a short-pulsed probe laser. Such pulsed laser matches the thermal decay of a nanoparticle, which significantly improves the sensitivity of wide-field MIP microscopy, enabling the fingerprint analysis of single bionanoparticles.
- 2) Defocus-enhanced MIP contrast: In dark-field MIP microscopy, the MIP contrast is proportional to the DC signal. We realize in the current work that this wisdom does not hold for interferometric MIP. By implementing piezo scanning in the z-direction, we developed defocus-enhanced wide-field (WIDE) MIP imaging. This novel approach allowed us to further enhance the MIP contrast and achieve high-throughput imaging of individual viruses.
- 3) Versatile applicability: While our initial focus was on beads and viruses, we demonstrated the versatility of our WIDE-MIP system by successfully imaging complex biological samples. We showcased the imaging of proteins and lipids in single mammalian cells, as well as the continuous mapping of protein distribution in live bacteria.

Through these instrumental advancements and novel concepts, we have successfully demonstrated rapid fingerprinting of small viruses (100-200 nm) with WIDE-MIP. We acknowledge that beads and viruses have relatively simpler structures compared to other biological samples such as cells. However, it is precisely due to their small size and unique structure that our previously published

instruments (Zhang et al., Sci Adv, Yin et al., Nat. Commun., Bai et al., Sci Adv., Zhao et al., Nat Commun.) lacked the sensitivity to achieve fingerprinting analysis of individual viruses. Furthermore, the WIDE-MIP system is also capable of analyzing the samples with complex structures or environment, which we have validated in our following response.

In our previous work (Bai Y, et al. Science advances, 2019, 5, eaav7127), we used a visible LED as the probe, which had a relatively long pulse duration ($\sim 1 \mu\text{s}$). This long pulse duration limited the applicability of MIP imaging to small nanoparticles, such as PMMA beads with diameters below 500 nm. The thermal decay constant of a single 200 nm PMMA beads after absorbing IR light was found to be 240 ns in air (Xia Q, et al, J. Phys. Chem. B 2022, 126, 43, 8597) and a short probe pulse is essential to probe such transient thermal effect. Thus, we replaced the LED light source with a pulsed nanosecond laser (NPL52C, Thorlabs, pulse duration of **129 ns**) to push the detection sensitivity to viral particle level.

It is commonly believed that when brightfield contrast is maximized, optimal MIP contrast is obtained, as the photothermal signal is modulated by the intensity of the visible light beam. While this wisdom holds for dark-field MIP microscopy, our work demonstrates that it is no longer the case for interferometric MIP detection of single bio-nanoparticles such as viruses. Instead, we achieved defocus-enhanced wide-field MIP imaging of single viruses by installing a piezo scanner (Piezosystemjena, MIPOS 100) on the objective, enabling precise z-axis scanning with a step size of 100 nm to realize defocus for interferometric scattering. This framework and setup provide guidance for maximizing photothermal signal on an interferometric (or iSCAT) microscope.

Moreover, our WIDE-MIP system is capable of analyzing samples with complex structures, such as cells and bacteria. As shown in Figure R1, we have demonstrated MIP imaging of proteins and lipids in single mammalian cells immersed in an aqueous environment.

Figure R1: WIDE-MIP imaging of mammalian cells cultured on silicon, cells were fixed and immersed in phosphate buffered saline (PBS).

Additionally, as shown in Figure R2, we also demonstrated the ability of WIDE-MIP to continuously map the dynamic distribution of proteins in live bacteria over 1 h. In this experiment, we introduced a heating plate to WIDE-MIP system to control the temperature of the bottom silicon substrate. This enabled time-lapse MIP imaging of proteins in live Escherichia coli, cultured in a TSB medium.

Figure R2: Time-lapse WIDE-MIP imaging of living *E. coli*: *E. coli* were cultured on silicon in TSB medium, temperature was maintained at 37 °C.

Together, these supplementary experiments demonstrate the versatility and applicability of WIDE-MIP for analyzing complex biological samples beyond beads and viruses.

In summary, our work introduces several new aspects in instrumentation and the concept of defocus-enhanced MIP contrast. While we agree with the reviewer about the importance of studying complex biological samples, it is of great significance to study bio-nanoparticles, such as viruses and extracellular vesicles, which are essential to basic scientific research and disease diagnosis. Due to the large scope of applications, we do not include other biological samples (e.g., larger cells) in this manuscript. We appreciate the valuable feedback, and we hope the above clarifications address the concerns.

Comment 1: In Fig3 a,b, slightly bright signal showing beads-like structures in the background of the image can be seen. The authors should explain this.

Author reply: We thank the reviewer for pointing out this question. The beads-like structures observed in the background of Fig3 a,b are caused by the background noise, which is generated by the system noise, including the illumination of a multimode fiber, fluctuation of the sample stage and variation of the visible laser. Details are discussed as follow.

As mentioned in our **reply to General comment**, we deployed a pulsed nanosecond laser as the probe light in WIDE-MIP. The short pulse width of the laser can lead to the production of speckle noise. To mitigate this effect, we employed a square-core multimode fiber (M97L02, Thorlabs) to reduce the speckle artifacts. However, the use of the multimode fiber for illumination does not result in a perfectly uniform illumination, leading to a non-uniform background.

To demonstrate that the observed particle-like backgrounds are due to background noise, we demonstrated the interferometric scattering imaging of a pure silicon substrate without any sample, along with its MIP image at an off-resonance wavenumber 1400 cm⁻¹ (Figure R3, please note that the black dots in the scattering image are stains on the camera). These images provide evidence that the particle-like backgrounds originate from pure background noise. We estimated the total fluctuation of the system, the standard deviation of the interferometric scattering imaging for 200 frames is ~ 9, which is contributed to the background noise. Despite the presence of slight background noise, WIDE-MIP still achieves a satisfactory signal-to-noise ratio (SNR) of about 21 for 100 nm PMMA particles (Supplementary Table 1 and Supplementary Fig. 3), which is acceptable for single-particle imaging.

Figure R3: Interferometric scattering and MIP imaging of silicon substrate.

Comment 2: Fig 4 shows comparison of fluorescence and MIP images. However, since the viruses are stained by GFP, I wonder the effect of GFP on MIP imaging. The discussion or explanation of the effect should be provided.

Author reply: We sincerely appreciate the valuable comment.

In this work, both VACV and VSV viruses were expressed with an enhanced green fluorescent protein (eGFP) envelope. The eGFP was fused to the VSV G protein, where each VSV contains ~ 1,200 molecules of the G protein on the viral surface (Journal of Virology, 2002, 76, 1309). With the formed G protein and G-eGFP fusion protein heterodimers, there are ~ 600 eGFP molecules on the surface of a single virus. Comparing the size of eGFP (2.4 × 4.2 nm) to that of the VSV virus (80 × 180 nm), we estimate that only 1.3% of a single VSV virus consists of eGFP. Thus, the effect of eGFP on MIP imaging should be negligible due to the relatively low content of eGFP in a single virus.

Supporting this notion, eGFP has a β-enriched sheet structure, while no obvious β sheet chemical signature was observed in either VACV or VSV expressing eGFP in Figure 4. Additionally, the pure VZV viruses without any labeling showed an enriched β sheet component. These findings further support that the effect of eGFP on MIP imaging is minimal.

Moreover, we note that the key distinction between DNA and RNA viruses lies in the different nucleic acid peaks associated with thymine (T) and uracil (U) residue vibrations, which is unrelated to the eGFP protein.

For the analysis of actual virus samples, label-free methods may be more suitable for diagnostic purposes. Thus, in this work, we firstly performed fluorescence-guided MIP analysis of single viruses by integrating fluorescence imaging and MIP imaging for accurate virus identification via WIDE-MIP (Figure 4). Subsequently, we performed MIP imaging and obtained fingerprint spectra of unlabeled pure VZV viruses to achieve label-free detection of single viruses (Figure 5). These approaches allow for comprehensive analysis while minimizing any potential influence of eGFP on the MIP imaging results.

We have added the above discussions to the revised manuscript.

Comment 3: Since the spectral resolution of MIP spectra is not sufficient to differentiate a few IR peaks overlapped with each other, fingerprinting viruses is ambiguous. The authors utilize QCL as an IR light source that could show the variation and the fluctuation of IR intensity. The intensity fluctuation might not be accurately compensated especially the effect could be dominated when the data point spacing is large. The authors should clarify that such low spectral resolution could identify the specific IR molecular vibrations in the spectra.

Author reply: We appreciate the reviewer's comment on the spectral resolution of MIP spectra. We agree that the limited spectral resolution may pose challenges in differentiating closely overlapped infrared peaks, leading to ambiguity in the identification of virus fingerprints. To address this concern, we demonstrate the stability of spectral identification and the methods used to differentiate different viruses in our study as follows.

Firstly, in our work, we utilized an OPO laser as the pump source for MIP imaging (M Squared Lasers). While the output power of the laser source can fluctuate, leading to variations in the detected MIP contrast, we have taken this into account during data acquisition and processing steps. To minimize the impact of power fluctuations, we employed a power meter fixed in the optical path to measure the real-time power at each wavenumber during the high-spectral-scanning. The raw spectrum was then normalized by the pump power.

To evaluate the spectral stability of WIDE-MIP, we selected the non-resonant region of PMMA vibrations from 1510 to 1610 cm^{-1} for analysis, which is the region used to acquire most of the data regarding viral particles. The standard deviation of the average MIP contrast within this range was found to be $\sim 0.16\%$, indicating stable high-spectral performance of WIDE-MIP in this spectral range (shown in Figure 3c or see the figure below). Additionally, the spectral fidelity was further confirmed by comparing the WIDE-MIP spectrum with FTIR absorption spectrum of PMMA. Therefore, readers should expect reliable data from this region for all samples. The revised discussion about the spectral noise and the Figure are shown as follows.

“The statistical spectra of 30 individual beads showed the distinguished resonance peaks of both C–H and C=O stretching vibrations (Fig. 3c, red line). The standard deviation of the mean MIP contrast within the range of ~ 1510 to 1610 cm^{-1} was found to be $\sim 0.16\%$, which corresponds to the off-resonance region of PMMA vibration. This demonstrates that the stable hyperspectral performance of WIDE-MIP. Furthermore, the spectral fidelity was confirmed by comparing the WIDE-MIP spectrum with FTIR absorption spectrum of PMMA (Fig. 3c, black line) ⁴².”

Fig. 3 Hyperspectral performance and spectral fidelity of WIDE-MIP microscopy. (c) MIP spectrum (red) and FTIR spectrum (black) of $D = 200 \text{ nm}$ PMMA beads. The averaged MIP spectrum was obtained from 30 single PMMA beads. Power before the objective: pump: 31.4 mW at 1452 cm^{-1} , 38.6 mW at 1728 cm^{-1} , probe: $\sim 1 \text{ mW}$. Image acquisition time: 2.36 s per wavenumber. The MIP spectrum was normalized by the IR power.”

Secondly, the differentiation of DNA and RNA viruses in this work was achieved by quantifying the MIP contrast of T and U residue peaks in VACV and VSV. As shown in Figure 4 below, the T residue and U residue vibrational peaks are located at 1580 cm^{-1} and 1640 cm^{-1} , respectively, with no overlap between these two peaks and a wavelength distance of 60 cm^{-1} . Although the MIP contrasts of VACV and VSV shows no statistically significant difference at 1580 cm^{-1} representing U residue (Fig. 4p, $P = 0.138892$), MIP contrasts of both T residue (Fig. 4q, $P = 0.000074$) and the ratio of T/U (Fig. 4r, $P < 0.000001$) show significant difference between VACV and VSV. This demonstrates that WIDE-MIP can effectively classify RNA and DNA viruses on their T and U residues. Thus, although the spectral

resolution may not be sufficient to resolve individual peaks, the distinctive overall patterns and characteristics in the MIP spectra enabled us to differentiate between different viruses.

Fig. 4 Fingerprinting detection of single VACV and VSV viruses. (m) MIP spectra of two single VACV and VSV viruses (blue and red arrows labeled). Statistical MIP spectra obtained from (n) 36 single VACV and (o) 33 VSV viruses. Power before the objective: pump: 22.9 mW at 1544 cm⁻¹, 29.1 mW at 1552 cm⁻¹, 34.5 mW at 1656 cm⁻¹, 35.8 mW at 1768 cm⁻¹, probe: ~1 mW. Image acquisition time: 2.36 s per wavenumber. The MIP spectrum is normalized by the IR power. Quantified MIP contrast of peaks at (p) T residue and (q) U residue of VACV and VSV. (r) Quantified MIP contrast ratio of peaks at T residue/U residue of VACV and VSV. Line, median; box, SD. Ns ($P \geq 0.05$) denotes no statistically significant difference. Asterisks **** ($P < 0.0001$) denotes statistically significant difference.

To further demonstrate the ability of WIDE-MIP to identify the specific IR molecular vibrations in single virus, we measured the FTIR spectrum of pure VZV powder for comparison with the statistical MIP spectra obtained from 30 single VZV viruses in WIDE-MIP (Figure R4). The results showed that WIDE-MIP can accurately identify VZV viruses and reveal T residue vibrations in viral nucleic acids, lipids and enriched β sheet components in VZV viral proteins, which are consistent with the biochemical components in VZV (details in the **reply to Comment 5**). Furthermore, the WIDE-MIP peak assignments of base residues in the nucleic acids were referred in literature (Appl. Spectrosc. Rev., 1970, 3, 45).

Figure R1: (a) Statistical MIP spectra obtained from 30 single VZV viruses. (b) FTIR spectrum of pure VZV virus powder.

Together, we appreciate the reviewer's suggestion and have added the FTIR spectra of PMMA nanoparticles and virus in the revised manuscript. We hope these clarifications adequately address the concerns about the spectral fidelity.

Comment 4: Related to Comment 3, complex IR spectra from biological samples are usually decomposed to accurately characterize vibrational modes of their biochemical components.

Unexpected changes of spectral features, such as peak splitting of the amide I band and shift of peak positions due to interactions of molecules and molecular states, were frequently seen in IR vibrational analysis of biological samples. Especially, IR spectra that the authors showed has the much lower spectral resolution than that in other literatures, which may result in misinterpretation of vibrational signatures in such complex IR spectra. The authors should state validness of their results on fingerprinting single viruses.

Author reply: We appreciate the reviewer's concern regarding the validity of our results on fingerprinting single viruses using MIP spectra with lower spectral resolution. We acknowledge that complex IR spectra from biological samples often require decomposition to accurately characterize the vibrational modes of their biochemical components. Interactions between molecules and molecular states can lead to unexpected changes in spectral features, such as peak splitting and peak position shifts in the amide I band.

In our study, we aimed to utilize MIP imaging as a rapid and label-free method for virus identification based on the overall spectral patterns and characteristics rather than relying on the precise assignment of individual peaks. While the spectral resolution of MIP spectra may be lower compared to other high-resolution techniques, such as traditional FTIR spectroscopy, we believe that the distinctive patterns and trends observed in the MIP spectra allow for differentiation of different viruses.

To address the concern raised by the reviewer, we have provided comparisons between the MIP spectra and FTIR spectra of VZV viruses to demonstrate the spectral fidelity and the ability of WIDE-MIP to identify the protein secondary structure in single viruses (detailed Figures and discussion shown in the **reply to Comment 3**).

Overall, the combination of MIP imaging and the analysis of distinctive spectral patterns can provide valuable information for the fingerprinting of single viruses, even with lower spectral resolution. We hope that these additional validations adequately address the concerns raised by the reviewer.

Comment 5: The authors claimed that vibrational peaks originating from amino acid residues could be seen in MIP spectra of viruses. However, I am not sure that the amino acid components were dominantly reflected on IR spectra of virus samples, in which many other compartments coexist, such as nucleocapsid proteins and envelopes (lipids, proteins). The authors should justify their MIP spectra exhibiting DNA/RNA compartments by discussing the number of biochemical components in viruses. In addition, FTIR spectra of the virus samples also verify the successful demonstration of fingerprinting of viruses.

Author reply: We appreciate the reviewer's comment and suggestion. We would like to clarify that in our work, we did not claim that any vibrational peaks originating from amino acid residues can be observed in the MIP spectra of viruses. We apologize for any confusion caused by the statement.

To address the justification of MIP spectra exhibiting DNA/RNA compartments, we have provided additional information on the biochemical components present in viruses. For a varicella-zoster virus (VZV), it has a lipid-rich envelope derived from cellular membranes, within which viral glycoproteins are inserted (Nat. Rev. Microbiol, 2014, 12, 197). Within the VZV, there are ~ 125-kb linear double-stranded DNA genome and ~ 3000 proteins (Nat. Microbiol., 2020, 5, 1542-1552). Notably, three envelope proteins, namely glycoprotein B, glycoprotein H, and glycoprotein L, have been identified as essential VZV proteins forming the core fusion complex (Nat. Rev. Microbiol, 2014, 12, 197). These proteins have known 3D structure, and a significant proportion of their secondary structures consist of β -sheets, with turn structures also present (Nat. Struct. Biol., 2003, 10, 980).

To demonstrate the accuracy of MIP spectra for virus fingerprinting, we further provided the FTIR spectrum of pure VZV powder for comparison with the statistical MIP spectra obtained from 30 single

VZV viruses in WIDE-MIP (Figure R4). The results showed that WIDE-MIP can accurately identify VZV viruses and reveal thymine (T) residue vibrations in viral DNA, enriched β sheet components in VZV viral proteins and lipids, which are consistent with the biochemical components in VZV. Furthermore, the WIDE-MIP peak assignments of base residues in the nucleic acids and other components were referred in literature (Appl. Spectrosc. Rev., 1970, 3, 45, Appl. Spectrosc. Rev., 2008, 43, 134). As suggested by the reviewer, we have added this information to the revised manuscript.

Moreover, we have demonstrated the MIP detection of pure DNA and RNA film samples to validate the detection of viral nucleic acids in single viruses (Supplementary Fig. 8), which are also consistent with the MIP spectra of single viruses.

Supplementary Fig. 8 WIDE-MIP spectra of pure chemicals. WIDE-MIP spectra of (a) dried pure protein (BSA) film, (b) dried pure DNA film (cDNA of melanoma cell), and (c) dried pure RNA film (ssRNA of T24 cell). The MIP spectra are normalized by the IR power under each wavenumber.

Together, we appreciate the reviewer's suggestion and have added the FTIR spectrum of the viruses in the revised manuscript. We hope these clarifications adequately address the concerns.

Comment 6: The authors described the future of rapid diagnostic tools using WIDE-MIP that they demonstrated. This is nice view and highly demanded for the current world, however, it is difficult to see that this method could work for clinical samples of saliva or breath condensates, which have a lot of foreign substances without washing or purifying processes. Usually, such non-target substances in clinical samples either deteriorate the detection sensitivity of techniques or make it difficult to probe specific vibrational modes of target molecules. It is great perspective, but I feel it seems to be exaggerating a little bit.

Author reply: We appreciate the reviewer's valuable comment on the potential application of WIDE-MIP for diagnostic purposes. We agree that the current state of WIDE-MIP may not be suitable for diagnostic applications, which handles more complex samples and requires simple architectures. Thus, we have toned down our claims for utility of WIDE-MIP for diagnostic purposes in the revised manuscript.

However, we would like to highlight that there is still potential for utilizing WIDE-MIP in diagnostic applications. To address the concern about complex clinical samples, microfluidics and antibody modified substrate can be used for immobilizing specific viruses in complex media such as serum, based on previous study (ACS Nano, 2016, 10, 2827, Figure R5). The authors demonstrated that there were 4095 bound rVSV-ZEBOV viruses captured on the antibody spot sensor surface (diameter of 150 μm) within 16 min. This indicates that effective immobilization of detectable viruses can be achieved within a reasonable time frame.

Figure R5: Microfluidics for viral capture and imaging, adapted from Figure 2 in ACS Nano, 2016, 10, 2827.

As suggested by **Reviewer #3 in comment 6**, we have toned down our claims for utility of WIDE-MIP for diagnostic purposes and we agree that it is a good idea to use WIDE-MIP to perform quality control of viral vectors. We have revised the potential applications for WIDE-MIP in the discussion section as follows.

“Benefit from the compositional analysis of single viruses in a label-free manner, we envision WIDE-MIP as an alternative analysis tool for viral vectors used in gene therapy. Viral vectors, including adeno-associated viruses, adenoviruses, and lentiviruses, are increasingly used in gene therapy but pose challenges for quality control testing and characterization due to their complexity^{62, 63}. To ensure a safe, consistent, and high-quality product, accurate and rapid analytical assays are needed to monitor quality attributes. Sodium dodecyl-sulfate polyacrylamide gel electrophoresis, mass spectrometry, immunoblotting, enzyme-linked immunosorbent assay, polymerase chain reaction, or transmission electron microscopy are used to identify protein, genome and capsid content^{64, 65}, but these assays can be time-consuming and require pre-treatments or extraction. To address these limitations, we further demonstrated high-speed chemical imaging of single VACV by reducing the acquisition time to 0.32 s per image per wavenumber of single viruses and the SNR of one single VACV is ~ 4 within the field of view of 24 by 24 μm (Supplementary Fig. 10). With the ability to rapidly acquire fingerprints of single viruses, WIDE-MIP can provide insights into the quality control of viral vectors, such as their identity, purity, and stability⁶⁴ (details in Supplementary Note 8).”

Minor comments

Comment 7: In the introduction, the authors cited several papers on MIP microscopy, but most of them are from the same group. As far as I have recognized this fields, there are a few more groups that develop MIP microscopy for biological samples (Samolis et al., Biomed. Opt. Express, (2021)/ Kato et al., Anal. Sci., (2022)/ Lim et al., J. Phys. Chem. Lett., (2019)). Hence, it would be better to cite those literatures to show readers that MIP microscopy has been recently emerged and attracted many attentions from a broad spectrum of researchers.

Author reply: We sincerely appreciate the valuable comments, which greatly helps in the overall background of our work. We have added more discussions on broad applications for biological samples and added more references into the Introduction session in the revised manuscript.

“Since the first demonstration of 3D MIP imaging of living cells,²⁹ MIP microscopy has enabled broad applications in life science, ranging from individual bacteria³⁰, single cells^{31, 32, 33}, sliced tissues³⁴, to entire organisms³⁵.”

Comment 8: In this work, viruses were not measured in liquid condition, but ambient condition. This might give influence on physical and biochemical properties of viruses. Could the authors give comments on that?

Author reply: We appreciate the reviewer's comment on the measurement environment. The choice

to measure viruses in air was made based on the specific setup and principle of WIDE-MIP system. The simulation of interferometric defocus-enhanced MIP contrast was established in an air condition. While measuring viruses in a liquid environment could provide more direct relevance to real-world applications, there are certain technical challenges associated with liquid measurements of viral particles < ~200 nm in our current setup. Firstly, when switching to liquid environment, the refractive index of the environment changes, along with the scattered and reflected fields change. These changes will significantly alter the detected interferometric and MIP contrast, which giving a much more complex condition for simulation. In addition, the refractive index of the viral particles is similar to that of water or PBS, which makes it challenging to detect small viral particles without any field or substrate enhancement.

It is important to note that the current study primarily aimed to demonstrate the capabilities of WIDE-MIP for wide-field chemical imaging and characterization of individual viruses. In air conditions, the viruses are immobilized on the silicon substrate surface, allowing for imaging and analysis using WIDE-MIP. This immobilization provides stability and enables the detection and characterization of individual viruses. While the detection of dried viruses may not fully replicate the complexities of a liquid environment, they provide valuable insights into the imaging and analysis aspects of the technique.

We agree that further investigations considering liquid measurements would be beneficial to expand the scope and applicability of WIDE-MIP. In the future study, we will focus on adapting the WIDE-MIP system to enable measurements of small bionanoparticles in liquid conditions. This can be achieved through development of appropriate microfluidic systems that maintain the integrity and functionality of the viruses during measurement, as demonstrated in previous study (ACS Nano, 2016, 10, 2827). Furthermore, we can use a layered silicon substrate ($\text{Si}_3\text{N}_4/\text{Si}$ or SiO_2/Si) to increase interferometric contrast by reducing background reflection and enhancing the scattered field (Biomed. Opt. Express, 2017, 8, 2976).

We have added a discussion on this limitation and the potential for future research directions in the revised manuscript as follows to address the concerns raised by the reviewer.

“Future improvement for WIDE-MIP can focus on fingerprinting viruses or exosomes in liquid conditions, allowing for detecting biological nanoparticle samples in their natural states. This can be achieved by incorporating microfluidic systems¹⁸ and designed substrates⁴³ to capture viruses and enhance imaging contrast in liquid measurements, which will broaden the applicability of WIDE-MIP to real-world applications.”

Comment 9: In previous work from the authors, instead of scattering-based MIP imaging, MIP imaging based on temperature-modulated fluorescence intensity was demonstrated and they improved the sensitivity by two orders of magnitude. The authors may want to add discussion about the implementation of the technique or comments on that.

Author reply: We appreciate the reviewer's comment on our previous work involving MIP imaging based on temperature-modulated fluorescence intensity. While fluorescence-based MIP imaging offers a larger modulation depth, there are certain limitations and considerations in implementing it in our current study.

One of the key requirements for fluorescence-detected mid-infrared photothermal microscopy (F-MIP) is a robust and high-quality fluorescence signal. In this regard, it is essential to highlight the difference in fluorescence probe labeling quantity between previous study conducted by Yi Zhang et al. (J. Am. Chem. Soc., 2021, 143, 11490) and the current work. In Yi Zhang et al's work, high concentrations of commercial chemical dyes (10 μM Nile Red or Rhodamine) were used for labeling high-content biological components, such as proteins in cells. While in this work, both VACV and VSV viruses were

expressed with an enhanced green fluorescent protein (eGFP) envelope. The expressed eGFP was fused to the VSV G protein, where each VSV contains ~ 1,200 molecules of the G protein on the viral surface (Journal of Virology, 2002, 76, 1309). With the formed G protein and G-eGFP fusion protein heterodimers, there are ~ 600 eGFP molecules on the surface of a single virus. Comparing the size of eGFP (2.4 × 4.2 nm) to that of the VSV virus (80 × 180 nm), we estimate that only 1.3% of a single VSV virus consists of eGFP. Thus, due to the lower content of total fluorescence probes in this study, the resulting fluorescence intensity is significantly weaker compared to Yi Zhang et al's work. Consequently, the reduced fluorescence intensity poses a challenge when attempting to implement fluorescence-based MIP imaging of single viruses in this work.

To demonstrate it, we used the same camera (FLIR, Grasshopper3GS3-U3-51S5M) as in Yi Zhang et al's work to perform the fluorescence imaging of eGFP-VACVs, and tried the same parameters with a camera exposure time of 50 ms and a gain of 20 dB. However, these settings failed to capture the fluorescence of single viruses. As shown in the revised Supplementary Fig. 9, we increased the exposure time to 500 ms for imaging aggregated viruses, while using the exposure time of ~ 5 s and maximum gain setting for single viruses. Although the photobleaching of aggregated virus showed a similar level as Yi Zhang et al's work (< ~ 10%), severe photobleaching was observed in single viruses (> ~ 95%). This photobleaching of single viruses limits the detection of photothermal modulation and acquisition speed. Considering that the MIP signal relies on the difference in fluorescence intensity between the IR-on (hot) and IR-off (cold) states, this severe bleaching at the single-virus level further compromises the reliability of F-MIP.

On the other hand, scattering-based imaging offers certain advantages, particularly in terms of the photon budget. When compared to fluorescence imaging, scattering-based techniques allow for shorter camera exposure times, higher full well capacity, and reduced saturation issues. In Yi Zhang et al's work, a CMOS camera (FLIR, Grasshopper3GS3-U3-51S5M) with a full well depth of ~ 10,000 and a frame rate of 20 Hz was used for wide-field FMIP imaging. In this work, we employed a camera with a frame rate of 1270 Hz and a full well capacity of 2 million wells (Q-2HFW, Adimec). This choice ensured that sufficient probe photons were received at each pixel, enhancing the quality of the scattering signal. Additionally, the interferometric geometry further enhanced the weak scattering signal of single viruses.

Taking into account these limitations and technical considerations, the fluorescence-based MIP imaging is not suitable for our current study. Instead, we focused on fluorescence-guided MIP analysis of single viruses. In this work, we first collected and analyzed the fingerprint spectra of eGFP-virus samples (VACV and VSV) and performed co-localization of fluorescence imaging and MIP imaging (Figure 4) to demonstrate the accurate identification of viruses using the WIDE-MIP technique. However, for the analysis of actual virus samples, label-free methods may be more suitable for diagnostic purposes. Thus, we further performed MIP imaging and fingerprint spectra of unlabeled pure VZV viruses in Figure 5 to achieve label-free detection of single viruses.

We appreciate the reviewer's insightful comments and have added a discussion addressing these issues in the revised manuscript as follows and in Supplementary Note 7 in the revised supporting information.

"In comparison to recently reported fluorescence-detected MIP (F-MIP) microscopy^{60, 61}, WIDE-MIP offers a distinct advantage in detecting bio-nanoparticles with low levels of expressed fluorescence tags (Supplementary Fig. 9, details in Supplementary Note 7). Although the photobleaching of aggregated eGFP-VACVs showed a similar level in F-MIP⁶⁰ (< ~ 10%), severe photobleaching was observed in single VACVs (> ~ 95%). This photobleaching of single viruses limits the detection of photothermal modulation and acquisition speed. Considering that the MIP signal relies on the difference in fluorescence intensity between the IR-on (hot) and IR-off (cold) states, this severe bleaching at the single-virus level further compromises the reliability of F-MIP. Instead, we focused

on fluorescence-guided WIDE-MIP analysis, which enables label-free chemical imaging of single viruses.”

Supplementary Fig. 9 Photobleaching analysis of single VACVs. (a-c) Continuous fluorescence imaging of VACVs from frame 1 to frame 3. (d) Defocused interferometric scattering image of the same area in (a-c). Scale bars: 10 μm . Image acquisition time: 5 s/image. Fluorescence intensity of (e) aggregated viruses ($n = 15$) and (f) single viruses ($n = 20$) in the field of view.”

Reviewer #3

General comments: In this research article, Qing Xia et al. propose a novel microscopy method for non-destructive imaging, evaluation and characterization of viruses. Proposed technique involves immobilization of viruses on a silicon substrate and illumination with visible laser beam. When particles are illuminated with mid-infrared laser, the infrared absorption characteristics of the particle changes the scattering properties which modulates the scattered and detected visible signal. The work presented in this article is certainly novel. Authors have done a fair job capturing the previous work on this field and have successfully identified the original aspects of their work. Wide-field imaging of single viral particles and analysis of their photo-thermal spectrum individually is challenging because of the small size of these particles. However, authors argue that combination of widefield interferometric imaging enables setting up the detection plane to an optimum z-location so that MIP sensitivity is enhanced. Very interestingly, authors demonstrate spectral response to tunable mid-infrared illumination can be used to fingerprint the individual viruses by characterizing the biochemical contents. In one example, they can distinguish if the imaged viruses have RNA or DNA in them by identifying Uracil and Thymine signatures in the detected MIP spectrum. In another example, authors demonstrate this technique can be used to identify the protein secondary structures in the viruses. However, the paper can be improved in several ways.

Author reply: We thank the referee for reviewing our manuscript, affirming its novelty, clarity and comprehensibility, and for her/his suggestions on how to improve the manuscript. The provided comments and questions are responded point by point below.

Comment 1: I found it very confusing how the technique is explained in the methods section. It looks like all of Fig.2 is dedicated to convincing the reader that "interference contrast and MIP contrast peaks at different planes", but not clear how this information is later used. Where the MIP contrast peak occurs depends on the particle size, it would be set at a certain Z-height, and the photo-thermal data would be collected at that focal plane for particle characterization. So, in essence, one can do a Z-scan of MIP contrast, and leave it at maximum contrast point. It is confusing how the interferometric contrast and its dependence to Z-scan is used.

Author reply: We sincerely appreciate the valuable comments.

In our experiment, we first use the interferometric contrast (IR off) to locate the samples under the microscope. Once the particles are seen in the focal plane, we then turn on the IR laser and fine tune the defocus by a piezo scanner to maximize the MIP contrast. This process minimizes the IR irradiation to avoid thermal toxicity to the sample.

We have added a detailed description in the revised manuscript as follows. We hope this clarifies the method and addresses the concerns.

"To experimentally validate the mechanism of interferometric defocus-enhanced MIP, $D = 200$ nm PMMA beads on a silicon substrate were used. We first used the interferometric contrast (IR off) to locate the beads under the microscope. Once the beads were observed in the focal plane, we then turned on the IR laser to 1728 cm^{-1} , which corresponds to the acrylate carboxyl vibration (C=O stretching) in PMMA. To optimize the MIP contrast, we manually adjusted the defocus with a piezo scanner. Subsequently, a series of interferometric and MIP images of the PMMA beads were acquired by scanning the focal position of the objective lens."

Comment 2: Another confusing part is what the images actually represent. When we are looking at the photo-thermal images in the paper (i.e. Fig-1b, Fig-3a and 3b, etc), are these what is seen on the CMOS camera? Since the MIP signal is the modulation of visible signal by the photo-thermal effect, I

assume these images represent how the visible reflectance/scattering pattern changes, so maybe there was some subtraction of original (Cold) state. Maybe readers who are very familiar with the method will not encounter the similar confusion, but unless that's the case, the readers will likely not understand the method well.

Author reply: We thank for this suggestion. Yes, the interferometric images (cold or hot state) were seen on the CMOS camera. The MIP image was generated from the interferometric contrast difference between IR on (hot) and IR off (cold) states. We have added a detailed description in the Method section and revised the Supplementary Fig. 1 as follows for a detailed illustration of the system.

“Data processing. The interferometric images were captured at a camera shutter speed of 1270 Hz. The MIP images were obtained as the intensity difference between the hot and the sequential cold frame, at the speed of 635 frames/s.

Supplementary Fig. 1 WIDE-MIP microscope setup and signal synchronization. (a) Schematic of WIDE-MIP microscope. The IR pump beam was generated by a tunable (from 1400 to 1800 cm^{-1}) mid-IR laser operating at 20 kHz repetition rate with a ~ 20 ns pulse duration, which was further modulated by an optical chopper. The visible probe was provided with a pulsed 520 nm nanosecond laser with a pulse duration of 129 ns. The interferometric scattering was recorded by a 2 million well-depth camera. A delay pulse generator is used to synchronize the pump pulse, probe pulse and camera. PBS: Polarizing beam splitter, OAPM: Off-axis parabolic mirror, QWP: quarter-wave plate, CMOS: complementary metal-oxide semiconductor. (b) Illustration for the synchronization and data acquisition of WIDE-MIP microscopy. To synchronize and acquire data for WIDE-MIP microscopy, the delay pulse generator was triggered by the output signal from the nanosecond IR laser. The oscilloscope was used to monitor the IR and visible pulses through a Mercury-Cadmium-Telluride detector and a photodiode, respectively. The IR pulses were modulated to a 50% duty cycle by the optical chopper. The camera trigger signal delay was adjusted to capture both IR on (hot) and IR off (cold) frames. The MIP contrast was generated by the subtraction of hot and cold frames.”

Comment 3: One final recommendation for the methods section - the depth of focus for the imaging system needs to be provided, since results has a Z-dependence.

Author reply: We thank the reviewer’s suggestions. We have provided the depth of focus as 503 nm, measured from FWHM of the MIP contrast profile in WIDE-MIP image from $Z = -1 \mu\text{m}$ to $Z = 1 \mu\text{m}$. We have added the detailed information in the revised manuscript as follows.

“It should be noted that the depth of focus for MIP imaging is 503 nm and the spatial resolution is 417 nm, measured from a 200 nm PMMA particle in MIP image captured at the defocus plane of $Z = 0 \mu\text{m}$ (Supplementary Fig. 5, details in Supplementary Note 5).”

“Supplementary Note 5. Spatial resolution of WIDE-MIP imaging. In addition, WIDE-MIP system allows depth resolved measurement, the depth of focus for the imaging system is 503 nm, measured as FWHM of the MIP contrast profile in WIDE-MIP image from $Z = -1 \mu\text{m}$ to $Z = 1 \mu\text{m}$ (Supplementary Fig. 5d).

Supplementary Fig. 5 Spatial resolution of WIDE-MIP imaging. (d) Axial cross-sectional profiles of MIP image across the same bead (shown in Fig. 2g) at $Z = 0 \mu\text{m}$. The depth of focus for MIP imaging is 503 nm, calculated from the Gaussian fitted FWHM.”

Comment 4: I think the paper has solid scientific basis, but conclusions towards utilizing this tool for diagnostic applications may not be well suited. First of all, even though it is called to be a wide-field imaging tool, the field of view is maybe about 50μm wide. If the goal is to analyze a clinical sample, what would be the mass transport rate needed to immobilize detectable number of viruses on the surface and within a small square like this.

Author reply: We appreciate the reviewer's valuable comment on the potential application of WIDE-MIP for diagnostic purposes.

For the concern of field of view, we agree that a larger field of view is preferred when dealing with the clinical samples. In the current configuration of WIDE-MIP, the visible probe beam illuminates the full field of view ($> 100 \mu\text{m}$), while the IR pump beam is loosely focused, providing the effective imaging field of view ($\sim 50 \mu\text{m}$) due to limitations in IR power density. The OPO laser (Firefly-LW, M Squared Lasers) used in our current WIDE-MIP system provides an output pulse energy of $\sim 1 \mu\text{J}$ in the mid-IR range. To address this limitation, a more powerful IR laser with high energy and IR focusing optics with longer focal length can be used to increase the field of view. For example, an OPO laser model such as the NT373-XIR from EKSPLA could offer an output pulse energy of around 1 mJ in the mid-IR range, which is 1000 times stronger than the laser we currently used. This significant increase in energy would provide a field of view ~ 30 times larger than the current setup.

With the enhanced field of view, we can couple our device with a microfluidics for high-throughput analysis. Based on previous work (ACS Nano, 2016, 10, 2827), microfluidics and antibody modified substrate can be used for immobilizing specific viruses. The authors demonstrated that there were 4095 bound rVSV-ZEBOV viruses captured on the antibody spot sensor surface (diameter of $150 \mu\text{m}$) within 16 min. This indicates that effective immobilization of detectable viruses can be achieved within a reasonable time frame.

In conclusion, we appreciate the reviewer's input, which has helped us refine our paper and provide a more comprehensive outlook on the capabilities and limitations of the tool.

Comment 5: Even if the mass transport issue is resolved, it's not clear how the viruses are identified within a noisy background as in Figures 4a and 4b. For the data in Fig. 4, authors used GFP to identify the viruses from the background, which is of course not possible in a label-free diagnostic situation.

So, how would they be identified from the background? Is the MIP spectrum good enough to call there is a virus in any particular location or sample?

Author reply: We appreciate the reviewer's comment regarding the identification of viruses within a noisy background. As mentioned, it is indeed challenging to label viruses in a diagnostic situation. In this work, we addressed this by initially analyzing the fingerprint spectra of GFP-tagged virus samples (VACV and VSV) and performed co-localization of fluorescence imaging and MIP imaging in Figure 4 demonstrated the accurate identification of viruses using WIDE-MIP technology. To achieve label-free detection, we further demonstrated MIP imaging and fingerprinting of unlabeled pure VZV viruses in Figure 5. We also provided the FTIR spectrum of pure VZV powder for comparison with the statistical MIP spectra obtained from 30 single VZV viruses in WIDE-MIP (Figure R1). The results showed that WIDE-MIP can accurately identify VZV viruses in a noisy background and reveal an enriched β sheet components in VZV viral proteins.

Certainly, in diagnostic situation, the composition of clinical virus samples can be more complex, which poses challenges for the application of WIDE-MIP in diagnostic analysis at this stage. However, based on our previous research (ACS Nano, 2016, 10, 2827), we can use antibody modification for initial screening of specific viruses with microfluidics in complex solutions such as serum. Then, based on the known fingerprint spectra of pure viruses, we can distinguish different specific viruses from impurities in the sample, enabling the identification of complex virus samples within a noisy background.

Figure R1: (a) Statistical MIP spectra obtained from 30 single VZV viruses. (b) FTIR spectrum of pure VZV virus powder.

Comment 6: Diagnostic applications require simple architectures that can be robustly implemented in very different settings. On the other hand, research and development tools has the flexibility to be complex and costly. Considering development of viral vectors for a vast number of different therapeutic applications gained popularity in the recent year, I recommend the authors to look into applications such as analysis and quality control of viral vectors. The proposed technology is certainly novel and has strengths, but it may be more suitable for applications such as this one.

Author reply: We thank the reviewer for the valuable comment on the potential application of our technology in the analysis and quality control of viral vectors. Indeed, the proposed WIDE-MIP technology has the potential to be used in various fields including biomedicine. We appreciate the suggestion and will certainly explore the potential application of WIDE-MIP in this area. We believe that our technology can provide valuable insights for the analysis and quality control of viral vectors.

In addition, we agree that at this stage, WIDE-MIP may not be suitable for diagnostic applications, which includes more complex samples and requires simple architectures. We have toned down our claims for utility of WIDE-MIP for diagnostic purposes in the revised manuscript. Besides, viral vectors are increasingly being used in different therapeutic applications, and quality control of viral vectors is essential for their successful development.

Thus, we have revised the **Abstract** and added the potential applications for analysis and quality control of viral vectors in the **Discussion** section as follows to further highlight the versatility of WIDE-MIP.

“Abstract: *Together, these advances open a new avenue for compositional analysis of viral vectors and elucidating protein function in an assembled virion.”*

“Discussions: *Benefit from the compositional analysis of single viruses in a label-free manner, we envision WIDE-MIP as an alternative analysis tool for viral vectors used in gene therapy. Viral vectors, including adeno-associated viruses, adenoviruses, and lentiviruses, are increasingly used in gene therapy but pose challenges for quality control testing and characterization due to their complexity^{62, 63}. To ensure a safe, consistent, and high-quality product, accurate and rapid analytical assays are needed to monitor quality attributes. Sodium dodecyl-sulfate polyacrylamide gel electrophoresis, mass spectrometry, immunoblotting, enzyme-linked immunosorbent assay, polymerase chain reaction, or transmission electron microscopy are used to identify protein, genome and capsid content^{64, 65}, but these assays can be time-consuming and require pre-treatments or extraction. To address these limitations, we further demonstrated high-speed chemical imaging of single VACV by reducing the acquisition time to 0.32 s per image per wavenumber of single viruses and the SNR of one single VACV is ~ 4 within the field of view of 24 by 24 μm (Supplementary Fig. 10). With the ability to rapidly acquire fingerprints of single viruses, WIDE-MIP can provide insights into the quality control of viral vectors, such as their identity, purity, and stability⁶⁴ (details in Supplementary Note 8).”*

“Supplementary Note 8. Potential application for rapid quality control of viral vectors.

We further demonstrated high-speed chemical imaging of single VACV by reducing the acquisition time to 0.32 s per image per wavenumber of single viruses and the SNR of one single VACV is ~ 4 within the field of view (FoV) of 24 by 24 μm (Supplementary Fig. 10). In comparison, the image acquisition time of one single virus is 46.4 s at the FoV of ~2.3 by 2.3 μm in previous work¹⁵. Together, WIDE-MIP microscopy provides ~1000-fold higher throughput, enabling fingerprinting of viral vectors for quality control.

To use this method, the viral vector products can be prepared on a silicon substrate following the sample preparation protocol and then imaged using WIDE-MIP. The substrate will then be taken for WIDE-MIP imaging and get the fingerprints of the particles from the products. For fingerprint region from 1500 cm^{-1} to 1750 cm^{-1} , fingerprinting at each field of view will take at least 8.32 s with the scanning step at 10 cm^{-1} . Quality control results can be obtained by comparing the fingerprints of the products with those from the standard viral vector, followed by spectral analysis of viral proteins and viral nuclei acids to determine stability, purity, and integrity. Additionally, WIDE-MIP can advance the development of new viral vectors by facilitating their characterization and optimization.”

REVIEWER COMMENTS

Reviewer #2 (Remarks to the Author):

The authors adequately addressed the majority of comments and suggestions, including mine, by presenting additional experimental results, which are indeed quite reasonable and valuable. Consequently, I am willing to consider accepting the manuscript. I have only one comment regarding the contrast between interferometric and defocused-enhanced MIP images. In this article, the authors have demonstrated that by intentionally inducing defocus, one can optimize the contrasts of wide-field MIP images. This implies that establishing a correlation between the contrast observed in MIP images and interferometric images of biological samples poses a challenge to correlate them to discuss about biological functions due to the "defocused" nature of the interferometric image. This defocus characteristic indicates an inaccurate representation of the shape and structure of the sample. Would it be possible for the authors to provide an elucidation on how the contrast disparities between MIP and interferometric images might affect the analysis of biological samples in particular, discussing on their biochemical and morphological alterations ?

Reviewer #3 (Remarks to the Author):

This study has some noteworthy results. Ability of generating information on nucleic acid and protein contents of individual viruses with a wide-field microscopy method is an important advancement in the field. The work is continuation of the previous work this group has performed, but certainly original as they show further improvements in detection method and support it with the data. It is certainly an advancement in wide-field optics and nano-particle / bio-particle / virus analysis.

The work somewhat loosely supports the overarching claims and conclusions. Provided data shows the potential for future applications, however it is too early to make a conclusion for its utility in diagnostic or quality control of viral vectors in real manufacturing settings. Authors will likely continue on demonstration of performance with real-world samples in future studies; however it is up to the editor if the current level of validation is sufficient for publication in Nature Communications.

The methodology and data analysis methods are very strong. Clarifications that were made after the initial review will certainly help the readers better understand the methods and data analysis. Provided level of data is adequate.

RESPONSE TO REVIEWERS' COMMENTS

Reviewer #2

General comments: The authors adequately addressed the majority of comments and suggestions, including mine, by presenting additional experimental results, which are indeed quite reasonable and valuable. Consequently, I am willing to consider accepting the manuscript.

Author reply: We thank the referee for his/her thoughtful evaluation of our manuscript and for considering accepting it. The provided comments are responded below.

Comment 1: I have only one comment regarding the contrast between interferometric and defocused-enhanced MIP images. In this article, the authors have demonstrated that by intentionally inducing defocus, one can optimize the contrasts of wide-field MIP images. This implies that establishing a correlation between the contrast observed in MIP images and interferometric images of biological samples poses a challenge to correlate them to discuss about biological functions due to the "defocused" nature of the interferometric image. This defocus characteristic indicates an inaccurate representation of the shape and structure of the sample. Would it be possible for the authors to provide an elucidation on how the contrast disparities between MIP and interferometric images might affect the analysis of biological samples in particular, discussing on their biochemical and morphological alterations?

Author reply: We appreciate the valuable comments from the reviewer. We want to clarify that the contrast disparities between MIP and interferometric images do not affect the analysis of biological samples, as our primary analysis is based on MIP contrasts. The interferometric images only serve to locate before acquiring MIP images.

In our experiment, the interferometric contrast is used to locate the samples under the microscope. Once the samples are seen in the focal plane, we then turn on the IR laser and fine tune the defocus by a piezo scanner to maximize the MIP contrast of the biological samples. Although the interferometric images may appear slightly defocused, it is crucial to note that the MIP images remain in focus, displaying distinct contrasts of the biological samples. As a result, MIP images not only provide valuable shape and morphological information of the biological samples, but also offer insightful biochemical details about their specific contents.

It is important to emphasize that defocusing interferometric image does not compromise the accuracy of the MIP imaging and its ability to discern biochemical and morphological alterations in the samples. On the contrary, the optimization of MIP contrast through defocusing enhances the sensitivity of our method, enabling us to obtain precise information about the analyzed biological samples.

We appreciate the reviewer's insightful comments and have added a discussion addressing these issues in the revised manuscript.

Reviewer #3

General comments: This study has some noteworthy results. Ability of generating information on nucleic acid and protein contents of individual viruses with a wide-field microscopy method is an important advancement in the field. The work is continuation of the previous work this group has performed, but certainly original as they show further improvements in detection method and support it with the data. It is certainly an advancement in wide-field optics and nano-particle / bio-particle / virus analysis.

The work somewhat loosely supports the overarching claims and conclusions. Provided data shows the potential for future applications, however it is too early to make a conclusion for its utility in diagnostic or quality control of viral vectors in real manufacturing settings. Authors will likely continue on demonstration of performance with real-world samples in future studies; however it is up to the editor if the current level of validation is sufficient for publication in Nature Communications. The methodology and data analysis methods are very strong. Clarifications that were made after the initial review will certainly help the readers better understand the methods and data analysis. Provided level of data is adequate.

Author reply: We sincerely appreciate the reviewer's positive evaluation of our work and the recognition of the originality and advancements presented in our study, as well as the noteworthy results achieved in virus analysis.

We acknowledge that there is ongoing work required to advance WIDE-MIP towards its potential applications in diagnostic or quality control of viral vectors in real manufacturing settings. While our current study represents an important step towards these applications, it is crucial to fully establish the practical utility of our approach. Therefore, as part of our continuous efforts, we are committed to further demonstrating the performance of our method with real-world samples in future studies.

REVIEWERS' COMMENTS

Reviewer #2 (Remarks to the Author):

The authors have adequately addressed my comment.

RESPONSE TO REVIEWERS' COMMENTS

Reviewer #2

General comments: The authors have adequately addressed my comment.

Author reply: We appreciate all the valuable comments from the referee and thank for considering accepting it.